# Soil water retention and hydraulic conductivity measured in a wide saturation range

Tobias L. Hohenbrink[1,2], Conrad Jackisch[3,4], Wolfgang Durner[1], Kai Germer[1,5], Sascha C. Iden[1], Janis Kreiselmeier[6,7], Frederic Leuther[8,9], Johanna C. Metzger[10,11], Mahyar Naseri[1,5], Andre Peters[1]

[1] Institute of Geoecology, Soil Science & Soil Physics, TU Braunschweig, Braunschweig, 38106, Germany
[2] Deutscher Wetterdienst (DWD), Agrometeorological Research Center, Braunschweig, 38116, Germany
[3] Interdisciplinary Environmental Research Centre, TU Bergakademie Freiberg, Freiberg, 09599, Germany
[4] Institute for Water and River Basin Management, Chair of Hydrology, Karlsruhe Institute of Technology (KIT), Karlsruhe, 76131, Germany
[5] Thünen Institute of Agricultural Technology, Braunschweig, 38116, Germany
[6] Thünen Institute of Forest Ecosystems, Eberswalde, 16225, Germany
[7] Institute of Soil Science and Site Ecology, TU Dresden, Tharandt, 01737, Germany
[8] Helmholtz Centre for Environmental Research - UFZ, Department of Soil System Sciences, Halle (Saale), 06120, Germany
[9] Chair of Soil Physics, University of Bayreuth, 95447 Bayreuth, Germany
[10] Institute of Soil Science, Center for Earth System Research and Sustainability (CEN), Universität Hamburg, Hamburg, 20146, Germany
[11] Institute of Geoscience, Group of Ecohydrology, Friedrich Schiller University Jena, Jena, 07749, Germany

*Correspondence to*: Tobias L. Hohenbrink (t.hohenbrink@tu-braunschweig.de)

**Abstract.** Soil hydraulic properties (SHP), particularly soil water retention capacity and hydraulic conductivity of unsaturated soils, are among the key properties that determine the hydrological functioning of terrestrial systems. Some large collections of SHP, such as the UNSODA and HYPRES databases, already exist for more than two decades. They have provided an essential basis for many studies related to the critical zone. Today, sample-based SHP can be determined in a wider saturation range and with higher resolution by combining some recently developed laboratory methods. We provide 572 high-quality SHP data sets from undisturbed, mostly central European samples covering a wide range of soil texture, bulk density and organic carbon content. A consistent and rigorous quality filtering ensures that only trustworthy data sets are included. The data collection contains: (i) SHP data: soil water retention and hydraulic conductivity data, determined by the evaporation method and supplemented by retention data obtained by the dew point method and saturated conductivity measurements, (ii) basic soil data: particle size distribution determined by sedimentation analysis and wet sieving, bulk density and organic carbon content, as well as (iii) metadata including the coordinates of the sampling locations. In addition, for each data set, we provide soil hydraulic parameters for the widely used van Genuchten-Mualem model and for the more advanced Peters-Durner-Iden model. The data were originally collected to develop and test SHP models and associated pedotransfer functions. However, we expect that they will be very valuable for various other purposes such as simulation studies or correlation analyses of

different soil properties to study their causal relationships. The data is available under the DOI link https://doi.org/10.5880/fidgeo.2023.012 (Hohenbrink et al., 2023; the final DOI will be registered before publication, please use the review link meanwhile: https://dataservices.gfz-potsdam.de/panmetaworks/review/5c617cd2664ea4d03e81301b5bc2236f1948a3cf7eb9bad48da940524f0cbac0).

## 1 Introduction

A sound understanding of the hydrological functioning of variably saturated soils in the environmental cycles is important for numerous applications in agronomy, forestry, water management and other disciplines. The hydrological functioning of soils is controlled by the soil hydraulic properties (SHP), specifically the water retention and hydraulic conductivity characteristics. SHP models are essential to simulate water dynamics, solute transport and energy transfers in the vadose zone using water flow and transport equations. Such SHP models are empirical mathematical representations of the highly non-linear soil hydraulic curves, which are parameterised based on measured SHP data. In order to estimate SHP from more accessible information, pedotransfer functions relate SHP parameters to basic soil properties like soil texture, bulk density, and organic carbon content ($C_{\mathrm{org}}$) (Vereecken et al., 2010; Van Looy et al., 2017).

Since the early applications of SHP models in hydrological simulations in the 1980s, there is a demand for such parameters (Carsel and Parrish, 1988). Commonly, they are derived for specific SHP models (Vereecken et al., 2010), which is most often the van Genuchten-Mualem model (Van Genuchten, 1980; Mualem, 1976). Fitting non-linear SHP models to observed data and developing pedotransfer functions both require large data collections containing information about SHP measured over a large range of saturation in samples with various combinations of basic soil properties. Such data collections are commonly based on individual soil samples from various profiles.

Due to methodological restrictions, data for such applications were first limited to few points on the soil water retention curve using ceramic pressure plate extractors and pressure-controlled hydraulic conductivity (Brooks and Corey, 1964). Since the late 1990s, different data collections of SHP and associated basic soil properties have been compiled. They formed the basis to develop various pedotransfer functions. The freely available Unsaturated Soil Hydraulic Database (UNSODA) provided by the U.S. Department of Agriculture comprises nearly 800 SHP data sets from disturbed and undisturbed samples (Nemes et al., 2001). It includes measurements of retention and hydraulic conductivity with different coverage of the saturation range as well as basic soil properties, e.g. information on soil texture or bulk density. UNSODA was an important basis to develop ROSETTA (Schaap et al., 2001; Zhang and Schaap, 2017), which is the most established pedotransfer function to predict the parameters of the van Genuchten-Mualem SHP model.

Another prominent large collection of retention and hydraulic conductivity data is the database of the Hydraulic Properties of European Soils (HYPRES) (Wösten et al., 1999) and its further development as the European Hydropedological Data Inventory (EU- HYDI) (Weynants et al., 2013) which is unfortunately not freely available. There are a few more specific SHP data collections, e.g. the HYBRAS data describing Brazilian soils (Ottoni et al., 2018), and the collection by Schindler and Müller

(2017) which contains only data measured with the evaporation method (Peters and Durner, 2008; Schindler, 1980). Recently, Gupta et al. (2022) gathered published soil water retention data from 2,702 sites, prepared them for use in land surface modeling and made them openly accessible.

The existing databases have undoubtedly supported a large number of hydrological studies leading to important conclusions, but they still have some limitations and shortcomings. Often, SHP data only cover a small part of the naturally occurring range of soil saturation. Gupta et al. (2022) emphasised that in many cases the retention data series contain only a few pairs of data and lack information in the wet region close to full saturation. Measured saturated hydraulic conductivity ($K_{sat}$) is included in several data collections, but detailed information about the unsaturated hydraulic conductivity is still rare.

It is technically possible to create pedotransfer functions using only retention and $K_{sat}$ data (Assouline and Or, 2013) as has often been done in the past. However, in such cases the shape of the hydraulic conductivity curves is predicted only from the water retention curve and scaled to match $K_{sat}$. Hence, the absolute position of the conductivity curve is solely determined by a single $K_{sat}$ value, which is strongly influenced by soil structure and macropore connectivity, which are often not recorded nor assessed at the time of sampling.

A serious development and rigorous testing of full-range SHP models always requires measured unsaturated hydraulic conductivity data. Zhang et al. (2022) showed impressively how fast a supposedly large number of available SHP data sets can collapse, when they are filtered by predefined data requirements. They initially gathered 19,510 data sets from established data collections and first narrowed it down to 14,997 data sets describing undisturbed samples. They then extracted 1,801 lab measured data sets with information about both soil water retention and hydraulic conductivity. Finally, they extracted data sets with at least six retention and seven conductivity data pairs, each of which contained at least three data pairs close to saturation at matric heads larger than -20 cm. They ended up with 194 data sets accounting for only 1 % of the initial number.

Given the wide variability of naturally occurring soils, many pedotransfer functions are based on data collections that contain rather limited soil information. Weihermüller et al. (2021) showed that the choice of the pedotransfer function used in a soil hydrological model can have considerable effects on simulated water fluxes. The artificial neural network behind ROSETTA has been trained with 2,134 retention curves, 1,306 $K_{sat}$ values and 235 unsaturated conductivity curves (Schaap et al., 2001; Zhang and Schaap, 2017). Considering the wide use of ROSETTA with more than 1,840 citations (retrieved from Scopus on 02/07/2023) it becomes apparent that the specific characteristics of only 235 unsaturated hydraulic conductivity data sets have been propagated into a large number of applications and conclusions.

However, pedotransfer functions can only predict the SHP within the range covered by the training dataset. Furthermore, they tend to reflect the individual characteristics of the training data, which are most pronounced in case of small databases. To prevent such bottleneck effects, the basis for pedotransfer applications needs to be further diversified. This requires new and independent, quality-assured SHP data collections. With advanced measuring techniques becoming standard in many soil physical laboratories, it is now much easier to obtain experimental SHP data over a wider range of soil moisture and in the desired high quality.

In this paper, we present a collection of 572 new data sets of soil properties measured in soil samples (Hohenbrink et al., 2023) that are independent of existing databases. Each data set contains (i) SHP, and (ii) basic soil properties such as soil texture, bulk density and $C_{org}$. The SHP data meet high quality requirements since they have been determined by combining state-of-the-art laboratory techniques, i.e. the evaporation method (Peters and Durner, 2008; Schindler, 1980), the dewpoint potentiometry (Campbell et al., 2007), and separate $K_{sat}$ measurements. In addition, each dataset has undergone thorough quality control. The data collection covers a wide range of soil textures. Soil texture information is provided according to both the German (Ad-hoc-Arbeitsgruppe Boden, 2005) and the USDA classification systems (USDA, 1999). Within the silt and sand classes, we also provide the sub-classes coarse, medium and fine according to the German system.

In support of the FAIR principles (Wilkinson et al., 2016), we provide free access to the data for the development of SHP models and pedotransfer functions. We expect them to be valuable for a variety of purposes such as simulation studies and statistical analyses of various soil properties.

## 2 Materials and Methods

### 2.1 Data sources

A community initiative for collecting and sharing consistent SHP data was launched by researchers from the Division of Soil Science and Soil Physics at TU Braunschweig. Scientists from four other institutions participated by providing data measured in their laboratories. Most of the data had already been used to answer individual research questions at various research sites (Jackisch et al., 2017; Kreiselmeier et al., 2019, 2020; Leuther et al., 2019; Jackisch et al., 2020; Germer and Braun, 2019; Metzger et al., 2021). Some existing but yet unpublished data sets have been reviewed and integrated into the data collection, too. In addition, we systematically added data from sites with soil characteristics that were missing from the data collection. Such data were explicitly measured for this data collection.

To be included in the data collection, the data sets had to contain soil water retention and hydraulic conductivity data, measured in the laboratory by the evaporation method, preferably supplemented by dewpoint method data and also measurements of saturated hydraulic conductivity. The data sets also had to include information about soil texture, and bulk density, and preferably $C_{org}$. We have aimed to cover the data space of these basic soil properties as completely as possible. Therefore, we also included data sets that lacked some of the preferred information, when they added new combinations of basic soil properties to the data collection.

### 2.2 Soil samples

Each data set is based on one undisturbed soil sample taken in situ with metal cylinders. In 542 cases the sample volume was 250 $cm^3$, while 30 samples had a volume of 692 $cm^3$ as indicated in the metadata table of Hohenbrink et al. (2023). For the measurement of $C_{org}$, soil texture and retention data in the dry range (dewpoint method), disturbed soil (sub)samples were taken. In 363 cases, exactly one disturbed sample was assigned to each undisturbed sample, either by taking both samples in

close proximity to each other or by taking the disturbed sample directly from the undisturbed sample material after measuring the SHP. In the other 209 cases, the disturbed sample was taken as mixed material, representative for an entire site with several undisturbed sampling points. Consequently, in the latter cases the soil variables derived from the aggregated disturbed samples have been assigned to more than one data set (indicated in the metadata table). Information about the positions of the sampling sites is available for 555 data sets. It has either been measured by GPS or was taken from aerial images after sampling. The geo-positions are reported with a lateral accuracy of 100 m, which represents the best accuracy class in Gupta et al. (2022). The sampling depth is reported for 474 samples in the metadata table.

### 2.3 Laboratory measurements

Soil water retention in the wet (defined here as pF < 1.8; $pF = log_{10}(-h\ [cm])$) and medium (defined here as 1.8 < pF < 4.2) moisture range and hydraulic conductivity in the medium moisture range were simultaneously determined with the simplified evaporation method (Peters and Durner, 2008; Schindler, 1980) using the HYPROP device (METER Group, AG, Germany). The evaporation method provides information related to the drying branches of the SHP curves. The air entry points of the tensiometer cups were used as an additional measuring point (Schindler et al., 2010) in cases where the duration of the evaporation experiments was long enough. Soil water retention information was supplemented mainly in the dry moisture range (defined here as pF > 4.2) by measurements with the dewpoint method (Campbell et al., 2007; Kirste et al., 2019) using the WP4C device (METER Group, Inc., USA). Hydraulic conductivity of the saturated soil was measured in the undisturbed samples either with the falling head or the constant head method using the KSAT device (METER Group, AG, Germany). Particle size distributions of the disturbed soil samples were determined by wet sieving for the sand fractions and sedimentation methods for the silt fractions and clay content (DIN ISO 11277, 2002). The sedimentation analyses were carried out with slightly different approaches in each lab as specified for each dataset in the metadata table. The respective particle size classes were defined by the German soil classification system (Ad-hoc-Arbeitsgruppe Boden, 2005). Because the German system differs from international standards in the boundary between silt and sand (German: 63 µm, USDA: 50 µm) we additionally converted the texture data by interpolation with monotone cubic splines fitted to the cumulative particle size distributions as recommended by Nemes et al. (1999). Illustrations showing data in the texture triangle were created using the "soiltexture" R-package (Moeys, 2018). Bulk density of each sample was determined by oven-drying for at least 24 h after the evaporation experiments. $C_{org}$ was determined with high-temperature combustion using different elemental analysers, which are listed in the metadata table.

### 2.4 Data preparation and quality check

The results of all SHP measurements have been compiled with the HYPROP-FIT software (Pertassek et al., 2015). It was developed to organise and evaluate raw data from the simplified evaporation method, the dewpoint potentiometry and individual $K_{sat}$ measurements.

Despite a high level of automation and standardisation, manual adjustments to selecting the raw data for evaluation is required. To avoid misalignment due to differences in the manual treatment, all resulting retention and hydraulic conductivity points have been re-checked for plausibility by the same expert based on the following procedure:

1. Tensiometer check and offset correction: HYPROP uses two tensiometers at different levels. If in the first hours of the experiments (close to saturation) the measured difference between the upper and lower tensiometers deviate from the actual difference of 2.5 cm by more than 1 cm, an offset correction was performed to prevent unrealistic hydraulic gradients during data evaluation.
2. Consistency check if the initial water content was smaller than the porosity: If not, a slightly larger column height (1 - 4 mm) has been assumed to account for surplus water in the data evaluation.
3. Setting the evaluation limits of the evaporation method: Because not all measurements follow idealistic conditions, the data for evaluation have been limited to plausible records (capillary connection of the tensiometers, plausible upward gradient, omission of scattered values for unsaturated conductivity near saturation).
4. Omit retention data of the dewpoint potentiometry outside its validity limits: dewpoint potentiometry measurements tend to be less precise for lower tensions. To avoid unnecessary variance between the different methods (dewpoint and evaporation), values below pF 4 were omitted.
5. Plausibility of hydraulic conductivity values: In cases of values for unsaturated conductivity exceeding the separately measured saturated conductivity, such values were omitted.
6. Visual alignment check for data from the three methods ($K_{sat}$, evaporation, dewpoint) and omission of obviously misaligned datasets from the collection.

The original binary HYPROP-FIT files are provided by Hohenbrink et al. (2023) to ensure transparency on all manual adjustments. The final series of measured retention and hydraulic conductivity data were exported from HYPROP-FIT to csv-files for further data processing, which was mainly performed in R (R Core Team, 2020).

## 2.5 Fitting models to measured SHP data

For direct access to resulting SHP model parameters, we fitted two models to the measured soil water retention and hydraulic conductivity data using a shuffled complex evolution (Duan et al., 1992) in SHYPFIT 2.0 (Peters and Durner, 2015). The first model is the well-established van Genuchten-Mualem (VGM) model (Van Genuchten, 1980; Mualem, 1976). The second model is the recent version of the Peters-Durner-Iden (PDI) model with the VGM model as the basic function (Peters et al., 2021, 2023).

The PDI model specifically considers (i) capillary water in completely filled pores and (ii) non-capillary water in thin films on particle surfaces and in corners and ducts of the pore system. The explicit consideration of non-capillary water yields more realistic retention and hydraulic conductivity curves in the medium and dry moisture range. Furthermore, the description of hydraulic conductivity in the dry range includes an effective component that reflects isothermal vapour flux (Peters, 2013).

Retention and conductivity parameters were estimated simultaneously. During model fitting the few retention points measured with the dewpoint method were weighted ten times higher than those obtained with the evaporation method because the latter have a much higher abundance. Weights of hydraulic conductivity data were defined in a way that their ratio to the mean retention data weights was 16 to 10,000 following Peters (2013). We neglected measured $K_{sat}$ values in the parameter optimization process, since they mainly reflect effects of soil structure (Weynants et al., 2009), which is not considered in the unimodal SHP models. The saturated hydraulic conductivity model parameter $K_s$ equals the hydraulic conductivity of the saturated bulk soil excluding the soil macropore network.

For the VGM model six parameters were estimated (residual and saturation water content $\theta_r$ (-) and $\theta_s$ (-), the shape parameters $\alpha$ (cm$^{-1}$) and $n$ (-), the tortuosity parameter $\lambda$ (-), and the saturated hydraulic conductivity parameter $K_s$ (cm d$^{-1}$)). The predefined parameter limits are listed in Table 1. The upper limits for $\theta_r$ and $\theta_s$ were defined as a fraction of porosity $\Phi$ to ensure physical consistency. For the PDI model, five parameters ($\theta_r$, $\theta_s$, $\alpha$, $n$ and $\lambda$) were estimated with the same settings and fitting algorithms as in Peters et al. (2023).

Unlike VGM and common models of SHP, where the relative hydraulic conductivity curve is scaled by the saturated conductivity $K_s$, the new PDI model structure allows to realistically predict conductivity data close to saturation, which are usually not available (Peters et al., 2023). To avoid an unrealistically sharp drop of the conductivity curve close to saturation for soils with wide pore size distribution, we constrained the conductivity model by a maximum pore radius (maximum tension) close to saturation with the "h-clip approach" (Iden et al., 2015). According to Jarvis (2007), the maximum tension was set to -6 cm (5 mm equivalent pore diameter). The saturated conductivity is defined as the predicted absolute conductivity at this tension. We refer to Peters et al. (2021, 2023) for a more detailed description of the applied version of the PDI model.

**Table 1: Upper and lower parameter boundaries for fitting the van Genuchten-Mualem model (VGM) and the Peters-Durner-Iden model (PDI). α and n: shape parameters, $\theta_r$ and $\theta_s$: residual and saturation water content, Ks: saturated hydraulic conductivity parameter, λ: tortuosity parameter. Note that the parameter boundaries for $\theta_r$ and $\theta_s$ are defined individually as a fraction of the porosity $\Phi$. The boundaries for $\theta_r$ and $\lambda$ differ between both models to ensure physical consistency. The lower $\lambda$ constraint for VGM is set to guarantee physical consistency while allowing for maximum flexibility.**

| | VGM | | PDI | |
|---|---|---|---|---|
| | lower | upper | lower | upper |
| $\alpha$ (cm$^{-1}$) | $10^{-5}$ | 0.5 | $10^{-5}$ | 0.5 |
| $n$ (-) | 1.01 | 8.00 | 1.01 | 8.00 |
| $\theta_r$ (-) | 0.0 | $0.25 \cdot \Phi$ | 0.0 | $0.75 \cdot \Phi$ |
| $\theta_s$ (-) | 0.2 | $\Phi$ | 0.2 | $\Phi$ |

| | | | | |
|---|---|---|---|---|
| $K_s$ (cm $d^{-1}$) | $10^{-2}$ | $10^{5}$ | - | - |
| $\lambda$ (-) | $1-\frac{2}{1-\frac{1}{n}}$ | 10 | -1 | 10 |

## 3 Data description

The data collection is structured in the following sections: (i) metadata (file: MetaData.csv), (ii) basic soil properties (BasicProp.csv), and (iii) SHP including measured points of the retention curve and hydraulic conductivity curve (RetMeas.csv, CondMeas.csv), (iv) optimized parameter sets for two SHP models (Param.csv) and (v) data series resulting from both SHP models (HydCurves.csv). Each dataset is labelled by a unique Sample-ID for easy joining of the different tables.

### 3.1 Metadata

The metadata table summarizes relevant information about the availability of the single variables in each data set. All 572 datasets contain SHP measurements by the evaporation method, 499 contain at least one dew point measurement and 409 data sets include $K_{sat}$ measurements (Table 2). In 370 data sets all three kinds of SHP information are available. In case of the basic soil properties, soil texture and bulk density are available for all datasets and $C_{org}$ is available in 488 cases. Complete information about all variables (SHP and basic soil properties) is contained in 315 data sets (57 %).

The data collection contains location information for 555 data sets (see Appendix Figure A1). The sampling sites are not arranged systematically, as the region of sampling has not been a criterion for data collection. They are rather clustered in the regions where the contributing groups have performed field work. Most of the samples have been taken in Central Europe ($n$ = 508). Few data sets come from Canada ($n$ = 29), Japan ($n$ = 5) and Israel ($n$ = 30).

**Table 2: Key variables contained in the data collection, laboratory method used for analyses and number of available samples.**

| Measured variable | Laboratory method | Number of available samples |
|---|---|---|
| Hydraulic Properties of unsaturated soil | Evaporation method (Peters and Durner, 2008; Schindler, 1980) using the HYPROP device (Pertassek et al., 2015) | 572 |

| | | |
|---|---|---|
| | Added measurements by air entry point of tensiometer (Schindler et al., 2010) | 286 |
| | Added retention measurements by dew point method (Campbell et al., 2007) | 499 |
| Hydraulic conductivity of saturated soil | Falling head or constant head method using the KSAT device (METER Group AG, n.d.) | 409 |
| Bulk density | Weight of oven dried (105°C) undisturbed samples (Dane and Topp, 2002) | 572 |
| Soil texture (63…2000 µm) | Wet sieving with 2000, 630, 200, 63 µm sieves (DIN ISO 11277, 2002) | 572 |
| Soil texture (≤ 63 µm) | Pipette method (Köhn, 1931) | 300 |
| | Pipette method (Moshrefi, 1993) | 78 |
| | Hydrometer method (Dane and Topp, 2002) | 52 |
| | Integral suspension pressure method (Durner et al., 2017, Durner and Iden, 2021) | 94 |
| | Method unknown | 48 |
| Soil organic carbon content | High-temperature combustion using different elemental analysers as listed in the metadata table | 488 |

### 3.2 Basic soil properties

The data collection covers a wide range of soil textures, including soils with up to 65 % clay and 93 % silt and 100 % sand (positions of symbols in the soil texture triangle, Figure 1). It covers the textures most frequently found in temperate climates. The main textural classes according to the German classification (Ad-hoc-Arbeitsgruppe Boden, 2005) account for 217 (sand), 146 (silt), 121 (loam) and 88 (clay) data sets (Figure 1a). The sandy soils are further subdivided into 39 samples for pure sand and 178 samples for loamy sand, as the SHP usually have the highest variation within the sand texture class. The two areas in the soil texture triangle with the lowest data coverage are sandy clay and sandy silt. Figure 1d shows the colour coded samples in the USDA texture triangle to provide orientation for international readers.

The bulk density of the samples varies between 0.37 g $cm^{-3}$ and 1.89 g $cm^{-3}$ with a median of 1.40 g $cm^{-3}$. High bulk density mainly occurs in sandy soils while silty clay soils are less dense (Figure 1b and 1e). In general, soil bulk density scatters across

the texture triangle, which is reflected by rather weak but significant Pearson correlations (p-value <0.05) between bulk density and the sand ($r = 0.41$), silt ($r = -0.24$) and clay ($r = -0.50$) contents, respectively.

$C_{org}$ in the samples ranges from 0.04 % to 19.26 % with a median of 1.44 %. The maximum values occur mainly in silty clay, loam and silty sand soils. Smaller $C_{org}$ values are associated with sand and silt soils (Figure 1c and 1f).

In addition to the standard soil texture classification by sand, silt and clay fractions, the subgroups for silt and sand (i.e. coarse sand, medium sand, fine sand, coarse silt, medium silt, and fine silt) are provided for the German classification system (Figure 2a). Most silt soils contain a maximum fraction of coarse silt (20 µm - 63 µm), while the loamy sands are mainly dominated by the fine sand fraction (63 µm - 200 µm). In contrast to the weak correlation between soil texture with $C_{org}$ and bulk density, the latter are negatively correlated to each other (r=-0.76; Figure 2b).

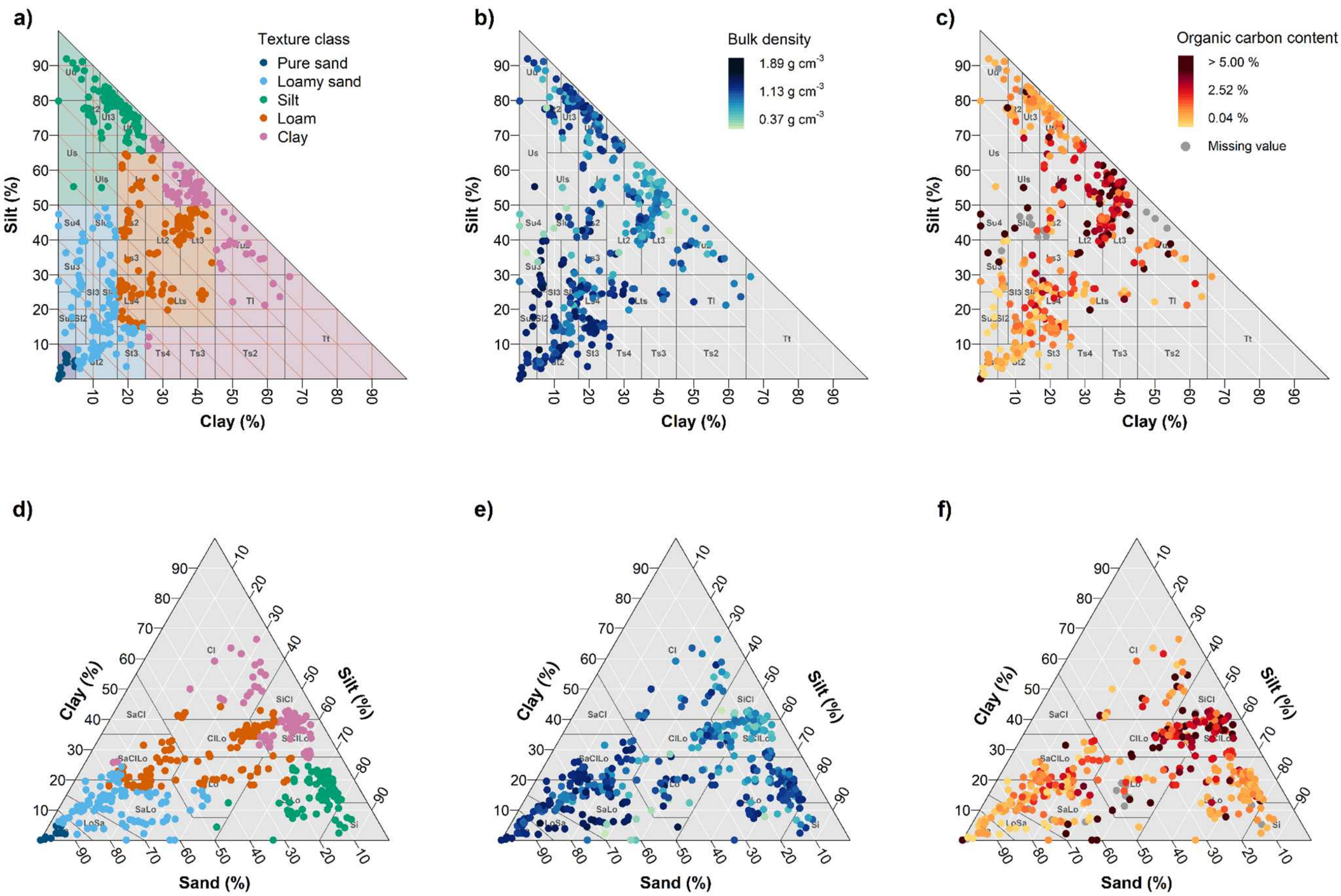

**Figure 1: Distributions of texture classes (a, d),bulk density (b, e) and organic carbon content (c, f) in the texture triangle of the German (a-c) and USDA (d-f) system.**

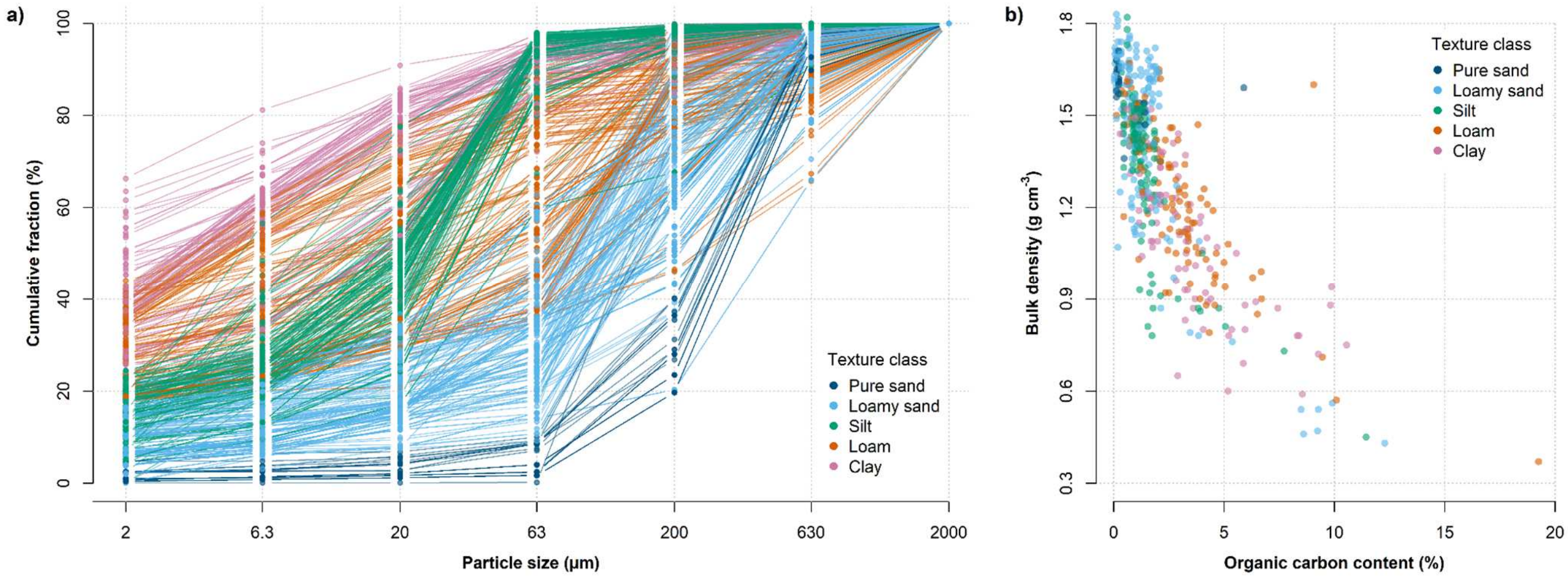

**Figure 2: a) Cumulative particle size distributions of the 572 samples (German classification system). b) Scatterplot of $C_{org}$ and bulk density (r=-0.76). The reference texture classes are colour coded.**

### 3.3 Measured soil hydraulic data

The measured SHP are shown in Figure 3. The retention data cover almost the entire range between full water saturation and oven dryness. The highest data coverage is available in the wet and medium saturation range with pF < 3.2, where the data stem from the simplified evaporation method. Half of the datasets contain one additional data point between pF 3.0 and pF 4.0 which originates from the air entry pressure of the porous tensiometers cup. In 499 data sets at least one data pair between pF 4.0 and pF 6.3 determined by the dewpoint method exists. To cover the drying branch towards pF 6.8 the number of measurements for single samples ranges between 1 and 8 (with a median of 3), because this method can only assess the matric head values after each reading of the respective sample states.

Hydraulic conductivity data obtained by the evaporation method range mostly from pF 1.0 to pF 3.2. Again, one single conductivity data point originates from the air-entry of the porous cup for about half of the datasets. A separately measured $K_{sat}$ is available for 409 datasets. The data collection neither contains conductivity data in the range close to saturation (pF <1), nor in the dry range. Currently, there is no standard laboratory method to determine hydraulic conductivity in this range. In the online version in Figure 3 the different methods contributing to the retention and conductivity data are plotted as circles (evaporation), triangles (air entry point), squares (dewpoint) and diamonds ($K_{sat}$). Figure 4 presents the same data colour coded by bulk density.

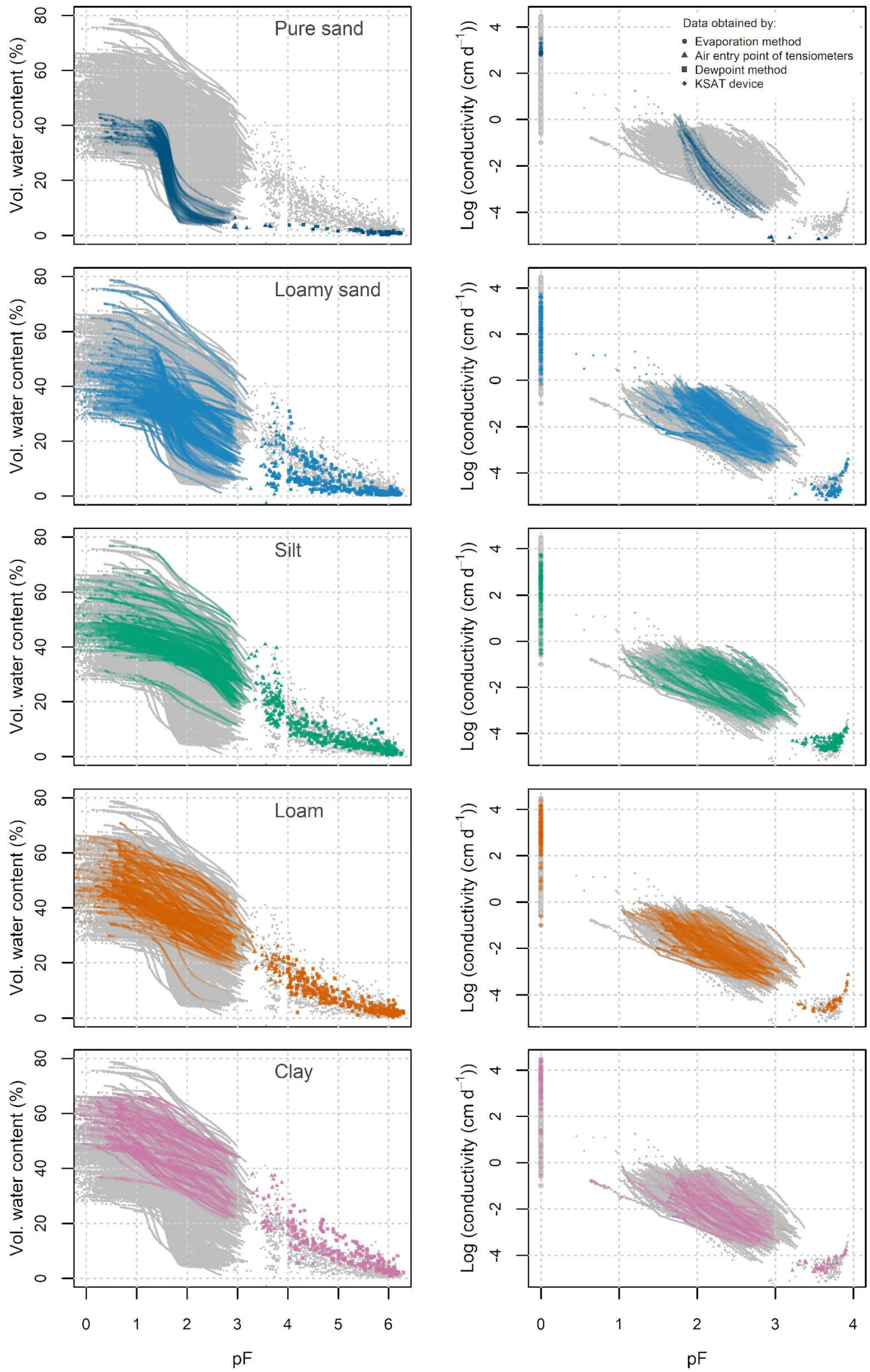

**Figure 3: Soil water retention (left) and hydraulic conductivity (right) grouped and colour coded by texture class. Grey background symbols show all measured values. Please note the different pF ranges for the retention and conductivity curves. In the online version the different laboratory methods contributing to the retention and conductivity data are plotted as circles (evaporation), triangles (air entry point), squares (dewpoint) and diamonds ($K_{sat}$) visible after zooming in.**

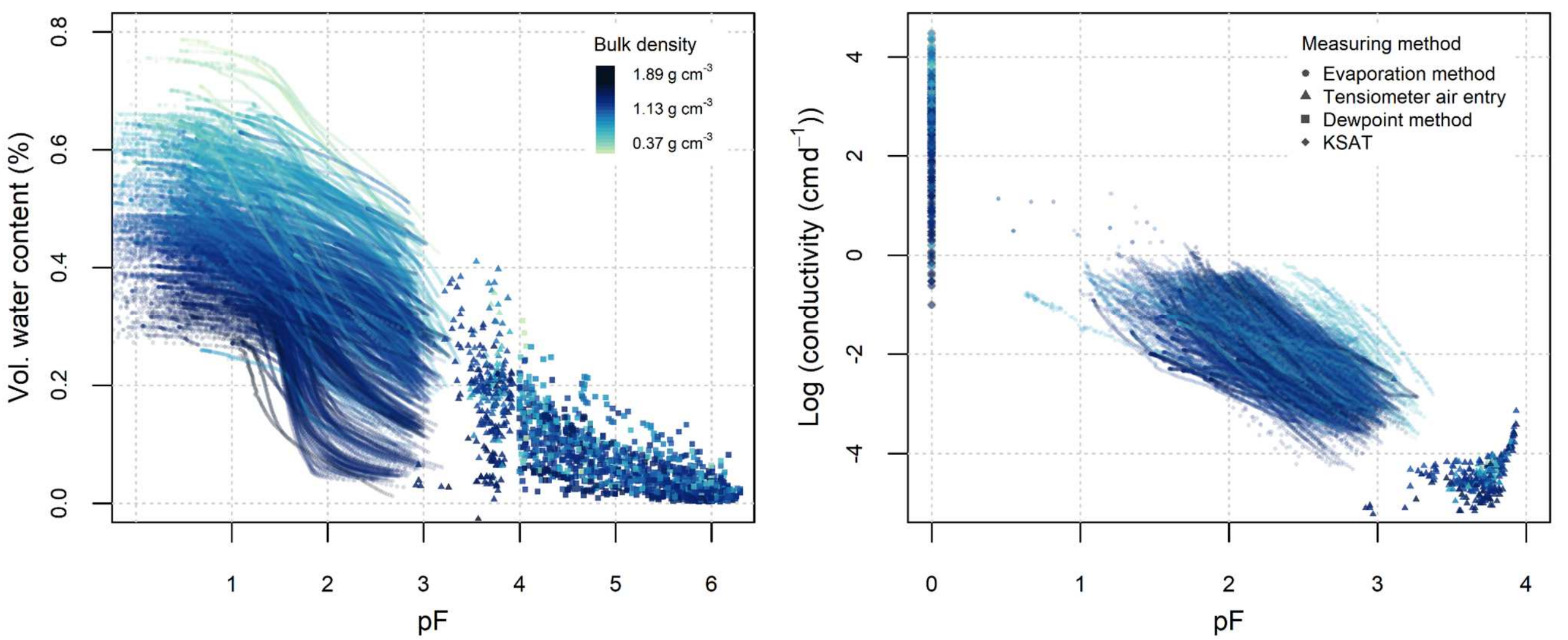

**Figure 4: Soil water retention (left) and hydraulic conductivity (right) colour coded by bulk density. Please note the different pF ranges for the retention and conductivity curves. A more detailed version with the used texture classes is in the Appendix Figure A2. In the online version the different laboratory methods contributing to the retention and conductivity data are plotted as circles (evaporation), triangles (air entry point), squares (dewpoint) and diamonds ($K_{sat}$) visible after zooming in.**

### 3.4 Fitted SHP models

The distributions of the fitted model parameters for both VGM and PDI (data provided in table "Param.csv" in Hohenbrink et al. (2023), but not shown here) mostly cover the predefined range of plausible parameters (Table 1). The constraining boundaries were only hit in 5 cases except for parameter $\lambda$ of the PDI model (154 cases in which bounds were hit).

The fitted water retention curves (Figure 5a and 5c) reflect the main characteristics of the measured SHP described above. Retention curves from both models are similar in the wet to medium range. However, in the medium to dry moisture range they systematically differ. The retention curves described by VGM approach a water content between 0% and 28%, while those from the PDI model consistently reach zero saturation at $pF = 6.8$, which reflects the matric potential at oven dryness (Schneider and Goss, 2012). The hydraulic conductivity curves described by VGM (Figure 5b) vary over a wide range. It proves difficult to visually relate the curves to respective texture classes, because the wet range part scales strongly with the existence of larger pores, and the shape of the curve is strongly limited by the underlying linear fit in log-log space. The PDI model curves (Figure 5d) are more closely related to texture and span a much narrower range for each texture class. The variation among the curves decreases towards the dry end of the saturation range. Especially in the dry range, the hydraulic conductivity increases along the texture gradient from pure sand via loamy sand, silt and loam to clay. This phenomenon results from the PDI model structure, where hydraulic conductivity in the dry range is directly derived from the water content at $pF = 5.0$ (Peters et al., 2021). At $pF > 5.5$ the hydraulic conductivity of the PDI model is dominated by the isothermal vapour conductivity for all texture classes.

As an estimate for soil water characteristics, we derived soil water content at field capacity (θ at $pF = 1.8$), soil water content at the permanent wilting point (θ at $pF = 4.2$), as well as the resulting plant available water content ($\theta(pF\ 1.8) - \theta(pF\ 4.2)$). Figure 6 shows these values in the texture triangle calculated based on the PDI retention curves. The water content at both, field capacity (Figure 6a) and wilting point (Figure 6b), roughly increases from sandy soils towards soils with finer textures. However, apart from this very general distinction, the values of both variables vary widely over the texture triangle, which directly results from the variation of the retention curves within a single texture class (Figure 5c). Plant available water content (Figure 6c) depicts the same high variability within the texture triangle. It varies between the extremes of 3.8 vol. % in pure sand up to 49.2 vol. % in fine-textured soil but does not align to any clear, texture-related pattern.

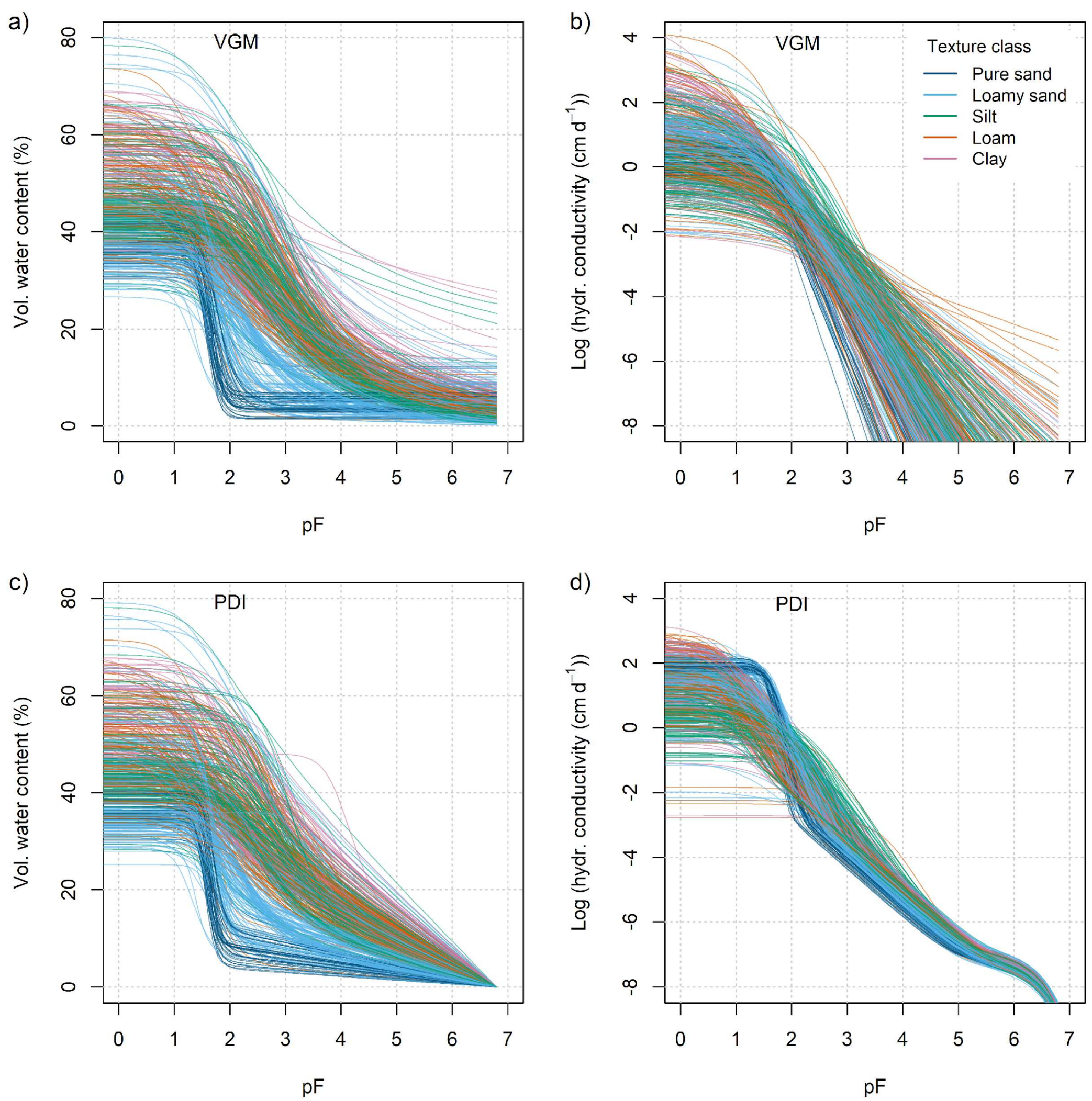

**Figure 5: Retention curves (left, a and c) and hydraulic conductivity curves (right, b and d) for the van Genuchten-Mualem model (a and b) and the PDI model with VGM basic function (c and d). Soil texture classes are colour coded.**

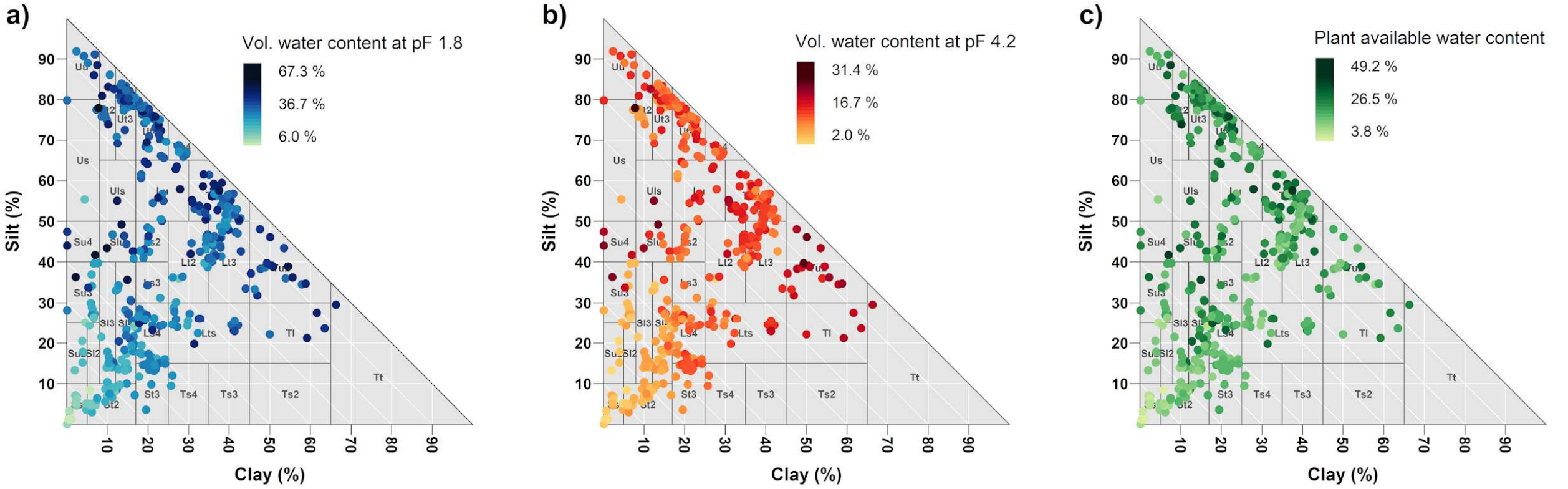

**Figure 6: Volumetric water contents at (a) field capacity, (b) the plant wilting point, and (c) the resulting plant available water scattered on the soil texture triangle. The values provided in the data collection are calculated from retention curves described by the PDI model.**

# 4 Discussion

## 4.1 New applications arising from the data collection

The combination of different state-of-the-art methods to measure soil water retention and hydraulic conductivity based on undisturbed samples yields a unique SHP data collection. Especially, the denser coverage of a wider range of saturation levels compared to existing data collections makes it valuable for new applications. For example, the retention data in the dry range measured with the dewpoint method represent essential information to develop retention models that overcome the concept of a residual water content, which has been shown to be not physically consistent (e.g. Schneider and Goss, 2012; Tuller and Or, 2005; Nimmo, 1991). Furthermore, this data collection provides measurements of both saturated and unsaturated hydraulic conductivity in high resolution and over a wide range of saturation levels. This supports the development and improvement of hydraulic conductivity models.

The high level of standardisation using the described methods enables us to link data from various labs without methodological offsets commonly found due to slightly deviating soil sample processing. Although the data set does not reach any global coverage, the data set exceeds existing SHP data collections by data density, extent of the values and variables, and consistency. We envision the proposed data structure as a foundation for upcoming additions with the methods becoming more and more accessible.

The VGM and PDI parameters provided in the data collection have been estimated with state-of-the-art techniques. Both parameter sets can be valuable to develop and test simulation models and to perform simulation studies. They can also serve as a benchmark for further developments of non-linear parameter estimation algorithms. We have intentionally omitted the measured $K_{sat}$ values during parameter estimation. Considering $K_{sat}$ in combination with unimodal SHP models usually causes

an overestimation of hydraulic conductivity close to saturation since the $K_{sat}$ information mainly reflects the impact of soil structure (Durner, 1994; Peters et al., 2023). The $K_s$ parameter derived for the PDI can thus be interpreted as the conductivity of the soil matrix only, excluding effects of soil structure, similarly to Weynants et al. (2009), and as further discussed in Fatichi et al. (2020). It is now possible to further investigate the relation between $K_s$ of the soil matrix and $K_{sat}$ of the entire soil including structure effects based on the parameters provided.

In addition to the commonly used three fractions sand, silt and clay to classify soil texture, we provide subgroups for sand and silt. Most pedotransfer functions only consider the three main texture groups as predictor variables. Our data suggests that the main texture classes alone contain limited information about soil hydrologic properties (large spread of hydraulic curves within texture classes in Figure 3, scattered texture distribution for plant available water in Figure 6c). Only in combination with bulk density and $C_{org}$, the data becomes more informative (Figure 4 and A2). For advancing pedotransfer functions, the presented data collection is a promising basis for analyses of (i) the resolution of texture data and texture class delineation (c.f. Twarakavi et al., 2010), (ii) the resolution of the SHP data series and (iii) suitable indicators for hydrologic functioning (field capacity, wilting point, etc. c.f. Assouline and Or, 2014).

### 4.2 Limitations of the data collection and further research needs

Although the data collection enables many different applications, it has some limitations that must be considered when analysing the data and interpreting the results. The single data sets are not completely statistically independent from each other for the following main reasons: (i) many samples stem from identical sites; (ii) some data sets exhibit identical texture and $C_{org}$ values, because in these cases only few aggregated disturbed samples representative for a whole site have been taken; (iii) the analyses have been performed in five different laboratories. However, by closely following the guidelines of the experiments a high degree of standardisation in the laboratory protocols could be achieved.

In some situations, it might be reasonable to thin out the data by keeping only data sets assumed to be statistically independent. However, whether this is necessary depends on the particular research question and the applied data analysing technique. We have decided to include all available data sets and to provide enough meta information to evaluate the statistical dependencies and leave it up to the user to decide how to handle such dependencies.

Another limitation of the data that users must cope with is the unbalanced distribution of the datasets in terms of basic soil properties. For example, Luvisols with silt contents between 70 % and 85 % are overrepresented in the data collection due to their agricultural importance which led to more frequent soil analyses. In contrast, there are data gaps for the sandy clay and sandy silt texture classes, because they do occur more sparsely in the regions under study and are generally less intensively investigated. The unbalanced distribution of the data can be especially challenging for the development of pedotransfer functions. This problem can be best solved by supplementing the data collection by additional measurements, but this is a major task at the level of the soil hydrological community and can hardly be achieved by individual researchers.

At the level of individual datasets, the gaps in the hydraulic conductivity series near saturation and under dry conditions are another important limitation. Such data are needed to parameterize existing models in a way that they become more reliable

in the respective saturation ranges. More comprehensive hydraulic conductivity data is also required to develop new SHP models. Therefore, we identify a need for developing and establishing new standard methods to measure hydraulic conductivity close to saturation (Sarkar et al., 2019b, a) and in the dry range.

### 4.3 Implications for lab procedures and further extension of the data collection

The different texture class definitions required us to estimate the missing breakpoint between USDA sand and silt fractions based on monotone cubic splines fitted to the German cumulative particle size distributions as recommended by Nemes et al. (1999). While this technique appears perfectly feasible given the high level of detail in the texture data with seven classes, this would have been more uncertain when the data would have been limited to the three main texture classes. The estimate can be eliminated altogether, when the 63 µm sieve and the 50 µm sieve are included as standard.

Despite the high level of standardisation using the described techniques to determine SHPs, the quality check based on the procedure presented in section 2.4 proved to be important to avoid erroneous interpretations and to ensure data quality. When followed, data from different labs can be easily combined. It would be favourable if this could be extended to further relevant soil properties e.g. soil texture, $C_{\mathrm{org}}$, Mid-Infrared reflection spectra. We encourage the community to use and extend this data collection.

## 5 Data availability

The data collection is hosted in the repository GFZ Data Services (Hohenbrink et al, 2023). It can be accessed via https://doi.org/10.5880/fidgeo.2023.012. The final DOI will be registered when the paper is accepted, temporary data access to Hohenbrink et al. (2023) via https://dataservices.gfz-potsdam.de/panmetaworks/review/5c617cd2664ea4d03e81301b5bc2236f1948a3cf7eb9bad48da940524f0cbac0/. The rights of use are defined by a creative common licence (CC BY 4.0). The data collection in the repository includes all data presented in this paper. Further information and materials such as small volumes of air-dried reserve soil samples can be provided by the corresponding author or the second author (Conrad.Jackisch@tbt.tu-freiberg.de) upon request.

## 6 Summary and conclusions

Motivated by a need for detailed soil water retention and hydraulic conductivity data, we collected data from 572 undisturbed ring samples in a community initiative. High level of standardisation in new measurement techniques and rigorous quality filtering allowed for consistency, which is rarely achieved in soil hydraulic analyses from different labs.

Initial comparisons of hydraulic indicators (e.g. plant available water content) with classical texture data showed very weak predictive power by texture. The addition of more texture classes from the particle size distribution and the addition of supplementary data on bulk density and organic carbon content appear to be more informative predictors.

The data collection can be used in its current form or integrated into existing data collections. All data sets were acquired directly from the original sources, which makes the data collection completely independent of the existing pool of data on SHP and thus contributes to their diversification. In particular, the hydraulic conductivity series will substantially expand the existing inventory of SHP data.

We expect that this data collection can serve as an independent, new and therefore unexplored benchmark reference to evaluate already existing SHP models and pedotransfer functions. Due to the high resolution of measured data compared to most data in existing databases and the extended range of saturation, it is also an ideal basis to develop and test new advanced SHP models and pedotransfer functions. It is well suited to verify findings and conclusions that have so far emerged from the existing data collections.

## Author contributions

TH compiled and analysed the data, created the figures, and drafted the manuscript in close collaboration with CJ. All co-authors contributed to the final version. JM, JK, FL, and CJ provided already existing data sets and evaluated them initially. KG and MN collected samples with new combinations of basic soil properties, performed laboratory measurements, and evaluated them initially. AP adapted the PDI model and the fitting software, was involved in building the data collection, and supervised the project. WD and SI supported the data preparation and analyses.

## Competing Interests

Conrad Jackisch is a member of the editorial board of Earth System Science Data. The authors have no other competing interests to declare.

## Acknowledgements

The initiative of this data collection has emerged from a project funded by the Deutsche Forschungsgemeinschaft (DFG, German Research Foundation grant PE 1912/4-1). TH and MN were funded by the same project. JM was funded by the Collaborative Research Centre "AquaDiva", funded as DFG SFB 1076, project number 218627073. CJ and TH were part of the DFG research unit "From Catchments as Organised Systems to Models Based on Functional Units" (FOR 1598) funded as DFG grant ZE 533/9-1. We thank Birgit Walter and Ines Andrae for laboratory analyses of the soil samples taken especially for this initiative. All authors thankfully acknowledge their respective field and lab support. Without their meticulous work this data set would not have come into existence.

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

## Appendix

Since global coverage and regional distribution of the sampling has not been a criterion for data collection, the samples are basically linked to the research activities of the contributors. Most samples have been taken in Central Europe (n = 508). A few data sets come from Canada (n = 29), Japan (n = 5) and Israel (n = 30) (Figure A1 for visual reference).

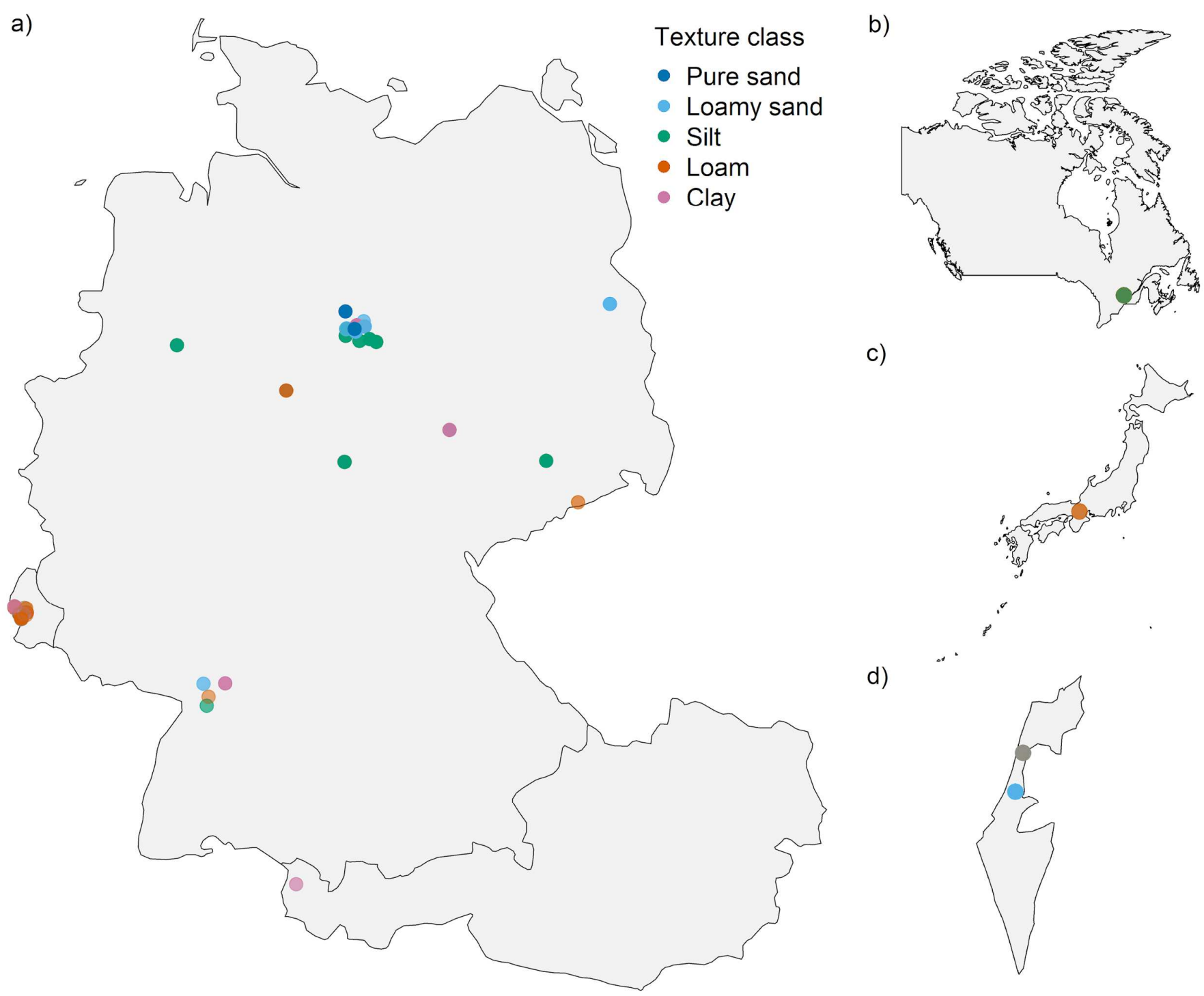

**Figure A1: Locations of the sampling sites in (a) Luxembourg, Germany and Austria, (b) Canada, (c) Japan, and (d) Israel. Please note that the map scales differ, as the maps should only provide a broad overview.**

Because soil texture classes did not provide strong information about the soil water retention and hydraulic conductivity curves, we have added the same plots as in Figure 3 colour-coded by bulk density (Figure A2).

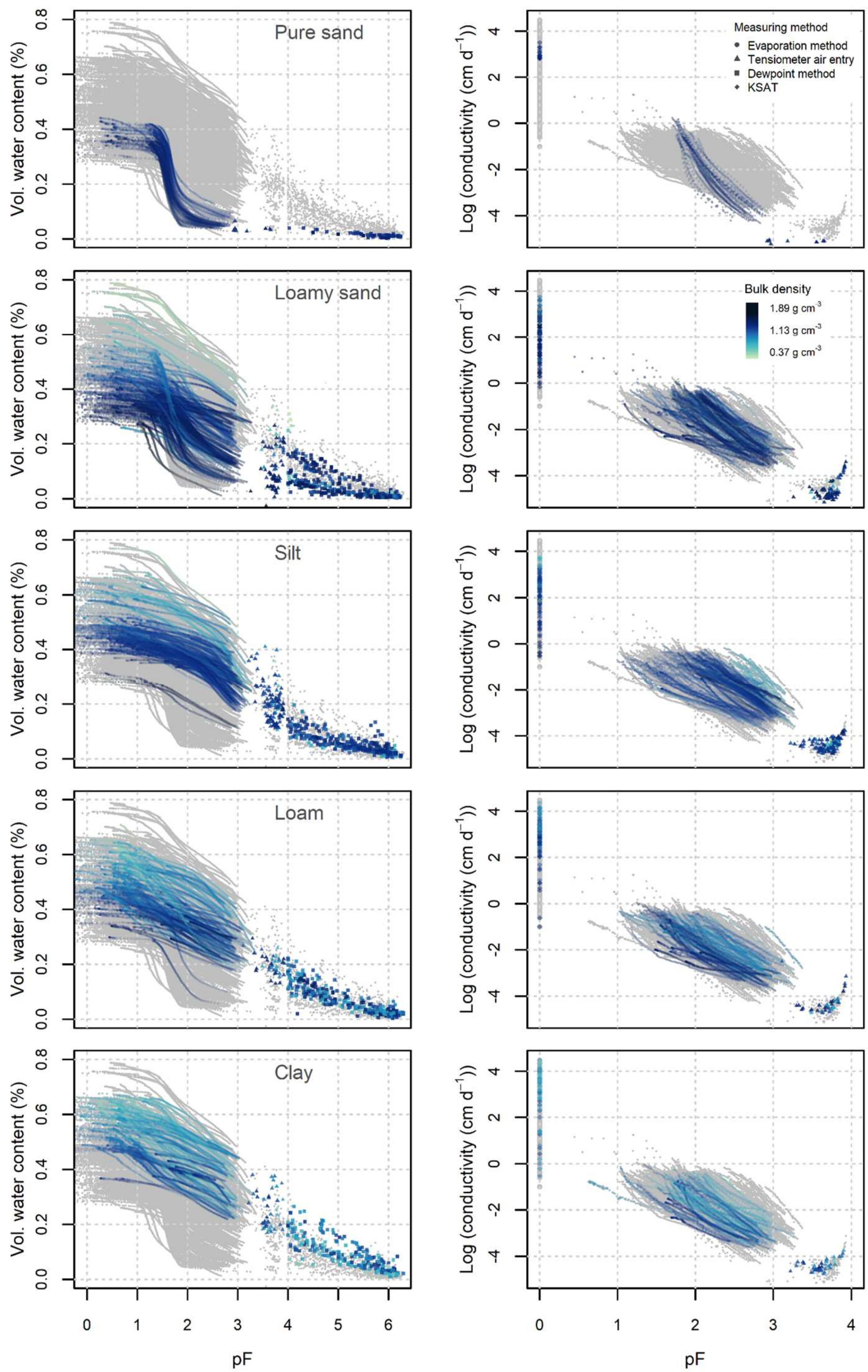

**Figure A2: Soil water retention (left) and hydraulic conductivity (right) colour coded by bulk density. Please note the different pF ranges for the retention and conductivity curves. In the online version the different methods contributing to the retention and conductivity data are plotted as circles (evaporation), triangles (air entry point), squares (dewpoint) and diamonds ($K_{sat}$).**