# Peer review of "Soil water retention and hydraulic conductivity measured in a wide"

_Earth System Science Data, 2023_

## Author Comment (AC1)

**Reply to Referee #1**

GENERAL COMMENTS

The presented dataset stores high quality soil physical data. The description of measurement methods and models applied to compute soil hydraulic parameters by fitting the moisture retention and hydraulic conductivity are detailed and clear. Structure of the manuscript is logical. The main strength of the database is the data on unsaturated hydraulic conductivity. This way the presented work and dataset will attain international interest.

The data could be easily accessed. Organization of the six data tables within the dataset is logical, the tables can be merged by the Sample_ID column.

Dear referee #1,
Thank you very much for your positive assessment and the thoughtful suggestions for a revision of our manuscript. We will address these line by line in the following.

A paragraph could be added about data quality check under materials and methods, because that could strengthen that the dataset was rigorously checked and the way the check was performed can be very informative for the readers and serve as a guideline.

You are perfectly right that such a quality check paragraph is a useful addition. It got lost during our initial internal revisions. We will add a respective paragraph before 2.4 to the manuscript describing the quality assessment of the data in more detail.

A final data check would be useful to secure that all data is correct. The detailed review can be found under SPECIFIC COMMENTS.

The data has been checked several times, but we will certainly perform another final check-up.

SPECIFIC COMMENTS

L24-25, L55, L78, L79, L82, L96 and entire text, please specify if you refer to soil profiles or soil samples, the word "data sets" is not enough specific.
We generally refer to soil samples throughout the manuscript. This will be specified more clearly. L24: We will add "sample-based SHP", L47: We will add the sentence: "Such data collections are commonly based on individual soil samples from various profiles.", For the examples, we will try to extract the requested differentiation from the cited literature.

L97: ... basic soil properties such as soil texture ... or something similar
Thank you for spotting this. We will amend the manuscript as suggested.

L101-102: please add reference or some examples for the two level texture information, because it is not widely used.
We thank the reviewer for this hint and will add the precise definition (particle sizes) of both the texture classes and the subclasses in the revised manuscript.

L127: ... mixed average soil sample ... is it correct?
Yes, the formulation was misleading. DISTURBED samples (originating from the undisturbed ones or having been sampled alongside the undisturbed samples) have been analyzed. In some cases, the disturbed reference samples have been referring to several undisturbed ring

samples. In such cases, the data were averaged and attributed to all rings. We will amend the formulation accordingly.

L134-149: all is clearly described, just a table providing an overview about the methods would be very informative, because for the readers it is a very valuable information what method was used for which soil property. Please add information about the measurement method of N and S, as well – because those are also included in the BasicProp.csv file. Please consider if the method used by soiltexture package can have limitations. Some other methods exist, which might result in a more accurate conversion to USDA silt and sand content. It is possible that in your case there would not be significant difference between different methods, but for other cases there might be. Readers might follow the procedure you published, so it worth to mention other methods, e.g.: Nemes et al (1999) https://doi.org/10.1016/S0016-7061(99)00014-2.
Thank you for your suggestion. We have discussed a more detailed table of all methods during the preparation. We will provide the table and to specify the methods. Please see the answer above for the texture class conversion.
We agree with the raised concerns about the log-linear interpolation in the "soiltexture" R-package. We will revise the conversion method according to Nemes et al. (1999) https://doi.org/10.1016/S0016-7061(99)00014-2 and Minasny and McBratney (2001) https://doi.org/10.1071/SR00065. We will revisit the data and use a more sophisticated method if possible.

L150: Before "2.4 Fitting models to measured data" subsection could you please add a separate subsection on how quality of the data was secured? Could you shortly describe what rules were applied during checking the data?
Thank you for your suggestion. And again, this has been discussed and will be included during the revisions.

L181-182: please add reference and equation used to compute parameter Ks of the PDI model.
The calculation of Ks was done as described by Peters et al. (2023), https://doi.org/10.5194/hess-27-1565-2023. We will make this clearer in the revised manuscript. Since this calculation procedure is rather complicated, we would like to restrain from a repetition of the procedure in this paper.

L184-186, Table 1: please add meaning of VGM and PDI to have the table self explanatory.
We will explain the abbreviations in the caption.

L190-193: would be informative to add 4.1-4.3 tables from 2023-012_Hohenbrink-et-al_Data-Description.pdf file here.
Thank you for your suggestion. We have discussed this question already and found the technical description alongside the data more suitable. However, we will consider a condensed version of the tables for more clarity here.

L194: It might worth to consider to create a metadata .xml file following the INSPIRE metadata guidelines (ISO 19115 and ISO 19139) and add it to the dataset.
We are aware of the different metadata guidelines and fully support the notion to emphasise their implementations. So far, we found an xml file somewhat repetitive to the tabled metadata. But we will consider your suggestion and revise for better compliance with the standards.

L216-219 and Figure 2. : please consider to provide this information according to USDA texture classes (based on the USDA sand, silt and clay fractions), because that is internationally used, the German texture classes are not widely known out of Germany. I see that for Figure 3. it might not make sense to use the USDA standard because than you might have only three fractions and Figures 4 and 5 is easier to interpret if meaning of texture classes can be read from Figure 3. Since we have done the reclassification and provide the USDA texture classes, it is very easy to use this reference and to switch the classification background. We will check this and include a statement in the figure description.

L241: circles on Figure 4 are hardly visible, maybe Figure 4 could be edited somehow to let easier distinguish between circle, triangle and square.
You are certainly challenging the capabilities of plotting so many data points without any aggregation obscuring the main point of the figure... Since all plotted values have been quality checked and since their origin does not really make a huge difference in the figure's interpretation, we have ended up with this hard to discern version. However, we will follow your suggestion (in line with referee #2) and try our best to find a new version of these plots.

L244: Please shortly add why number of dewpoint measurements ranges between 1 and 8.
We will revise the statement as follows: "Since the matric head results of the dewpoint measurements can only be assessed after each reading, the number of measurements for single samples ranges between 1 and 8 (with a median of 3) to cover the drying branch towards pF 6."

L263: ... range for coarser texture classes ... Do you agree?
No. Here we were not precise enough. We will rephrase the two sentences as follows: "The hydraulic conductivity curves described by VGM (Figure 5b) vary over a wide range and can hardly be grouped by texture classes. In contrast, those of the PDI model (Figure 5d) are more closely related to texture and span a much narrower range for each texture class. The reason for this behavior is that the conductivity curves must be extrapolated in the wet as well as in the dry moisture range and in the new PDI model formulation (Peters et al., 2021; 2023) this extrapolation is done on a physical basis."

L268-271: if th_1_8, th_2_5 and th_4_2 columns of Param table were computed with PDI model, please add "_PDI" as last characters to those column names.
Thank you for pointing to this. We will check the specific references and amend the document accordingly.

L272-273: please add very short explanation for why filed capacity and wilting point vary widely within texture classes. This is obvious for experts in soil physics but not that trivial for researchers from other environmental fields.
We will do so.

L308: please consider e.g. the work of Twarakavi et al. (2010) (https://doi.org/10.1029/2009WR007939 ) - or possible other papers in this topic – and rephrase the sentence accordingly.
Thank you for suggesting this citation and challenging our sloppy formulation. We will reconsider this argument and provide references.

L311: Do authors plan to add soil depth, chemical soil properties - e.g. pH or calcium-carbonate content - or taxonomical information to the dataset in the future? If soil depth is available it might be easy to add to the BasicProp.csv table, it could be an important data column.

This is true and will be done for soil depth. We will discuss what more we can include with sufficient degree of confidence. This issue arises with different standards for some of the analyses in the different labs. For the data we report, this has been checked and harmonised.

Result of checking the database:
there is a negative theta value in RetMeas.csv, please check and revise/correct.
there is a negative value for S in BasicProp.csv, please check and revise/correct.
Sum of USDA sand and silt and clay is 99.9 and 100.1 for some samples, it might worth to correct them to sum up to 100.

Thank you for pointing to these issues. We will carefully check the data again.

Dear referee #1,
Thank you again for your suggestions. We hope that we could address all concerns and will use your advice for a substantial revision of our manuscript and a check of the data.

Kind regards,
Tobias Hohenbrink, Conrad Jackisch, Wolfgang Durner, Kai Germer, Sascha Iden, Janis Kreiselmeier, Frederic Leuther, Johanna Metzger, Mahyar Naseri, and Andre Peters

---

## Author Comment (AC2)

**Reply to Referee #2**

The manuscript reports on a decently homogeneous data collection of soil physical and hydraulic properties that, in many aspects, provides more detailed data than what is available in most existing and internationally available databases. However, in certain aspects it provides less information, or potentially uses a weak solution for data harmonization. If unresolved, these can become its limitations, and eventually limit the anticipated benefit from reporting very detailed water retention data. This contribution of data is very welcome in the literature, but it would be desirable to report the data and some of the methods in more detail.

Dear referee #2,

Thank you very much for your critical assessment and the thoughtful suggestions for a revision of our manuscript. We will address these line by line in the following.

As a preamble to the forthcoming replies we are under the impression that some sort of misunderstanding underlies some of the critical comments. We seek to convey the combined soil water retention and hydraulic conductivity curve data as a valuable basis for a series of applications (including SHP model and PTF development among others). The additional data (e.g. organic carbon contents) has been added for further reference and as a "service" for a broader useability of the data. While your comments, suggestions and discussion aspects provide valuable hints for our revision, we also want to stick to our original plan for the publication and explanation of exactly this data. We will certainly elaborate more clearly on the respective methods, but we cannot perform additional analyses with respect to more texture data. Although we will revise the applied conversion of the texture classes, we are, of course, bound to the laboratory data we have. And again, we regard this data as additional information complementing the characteristic curves.

We sincerely hope that we have well understood your raised concerns and that our replies are capable of conveying our gratitude for your constructive criticism. We will ask the editor and the Copernicus office to extend the discussion phase to enable further exchange in the open discussion.

Three generic and a number of specific comments follow.
Some data, primarily particle size distribution data should be reported in more detail. I don't see trail of reporting more than sand-silt-clay contents, whereas the original set of measurements should be reported. There are emerging directions of research that would utilize that. It should also be communicated which of the silt-sand data pairs are from original measurements, and which have been a product of interpolation that adds additional noise.
We appreciate the suggestion and can do so. We focus on soil water retention and hydraulic conductivity curve data. We have tried to harmonize the data as much as possible and we did not anticipate such interest in these "auxiliary" data. Based on your comments, we will revise the data assemblage accordingly. However, unlike the highly standardized Ksat, Hyprop, WP4C analysis procedure, the texture analysis slightly differs between the originating labs and even for specific samples. Reporting the original set of measurements (as weight of retained sediment during wet sieving and sediment aspiration during sedimentation experiments including lab temperature etc.) is outside the scope of this publication and too much manual work to retrieve from the respective lab books. Since we are not aware of these "emerging

directions" of soil texture research, we would be pleased to include a respective statement and citation in the discussion to guide forthcoming versions of such datasets.

In addition, instead of the rather standard, flat data description, it is better value to go in depth on the exact steps that involved data manipulation, and present the outcome in a convincing way. I specifically refer to the particle-size conversions and its uncertainty, as well as the derivation of bulk density and return to these in the detailed comments section.
In line with referee #1 we agree that this part got shortened too much. Thank you for your comment. We will revise and extend the degree of detail in our methodological descriptions accordingly. However, we do not expect the main characteristics of the data to be dependent on the precision of the particle-size conversion. We report the texture data in 7 classes (German standards) all based on wet sieving and sedimentation analyses. (Few samples have been processed with Pario+, details will be added in the revisions.) In figures 4 and 6 we demonstrate that the information about the hydraulic functioning (retention and conductivity curves, PAW) is only weakly encoded in the texture information alone. This has been shown earlier too and underpins the main reason for us reporting the continuously measured unsaturated hydraulic conductivity in conjunction with soil water retention (although limited by methodological constraints).

Several statements made would have been true in the 1990s, but not anymore. I highlight some among the detailed comments. The authors should revise those and bring the statements up to standard according to the state-of-the-art in the 2020s.
We agree that we mostly refer to consolidated concepts in soil physics and soil analyses as we intend the dataset to be hopefully useful for a broader range of neighboring disciplines. We are under the impression that there is some misunderstanding due to awkward wordings on our side. We consider the SHP model PDI (as just published in a new version, Peters et al. 2023, https://doi.org/10.5194/hess-27-1565-2023) as up to date. With respect to the discussed usage of the data for PTF development, we refer to algorithms predicting SHP model parameters from more commonly accessible information in a continuous manner. Such PTF are driven by information about texture classes, bulk density and soil organic content.
Although we think that our arguments are not completely outdated, we will carefully revise them. Moreover we will clarify our wording to avoid any misunderstanding in the revised version.

Specific comments:
Abstract: Include a brief reference to the geographical extent (i.e. the contributing list of countries, with Germany dominating)
We will add "mostly central European" to L25.

L63: I believe Brazil has non-tropical data in HYBRAS as well. Please check and remove the word 'tropical' if necessary.
This is true and unnecessary. Tropical will be removed.

L66: replace 'commonly' with 'openly'
Will be done.

L72: ...as it has been often done historically.
Will be added.

L75: … which are often not recorded at the time of sampling.
"which are often not recorded nor assessed at the time of sampling" will be added.

L85-89: IT has been identified that ROSETTA's data is also gepgraphically skewed. It would be worth exploring where the 235 samples with unsatK come from.
Geographical skewness is an issue. However it is an issue which we hope to address in the future when we succeed in promoting the value of standardized analyses and evaluation as presented with the soil hydraulic property data. A review of the spatial references in the literature and its potential skewness is beyond the scope of our data manuscript.

L99: i.e. the evaporation
Thank you. Will be amended.

L99: dew-point potentiometry (Campbell et al., 2007)
Thank you. Revised.

L101: these are not two 'levels', but two standards. Use e.g.: "provided according to both the German and the USDA classification systems, and the…"
Thank you. Our wording was unclear. We will revise the sentence to "The information on soil texture is provided as main texture groups for the German and the USDA classification systems. Within the silt and sand classes, we also provide the sub-classes coarse, medium and fine according to the German system."

L103: please avoid using text like "strong foundation". The users and history will decide that.
L104-105: delete this sentence, it is repeated from earlier. Remove the dependence of the next sentence on this sentence (reference to further purposes).
L109: This is the 3rd mention of "various original purposes". Perhaps remove both earlier mentions.
All true. The statements will be shortened and rephrased.

L129: here you mean aggregated DISTURBED samples, is the correct?
Yes, the formulation was erroneous. Thank you for pointing this out. DISTURBED samples (originating from the undisturbed ones or having been sampled alongside the undisturbed samples) have been analyzed. In some cases, the disturbed reference samples have been referring to several undisturbed ring samples. In such cases, the data were averaged and attributed to all rings. We will correct the formulation accordingly. Information about which undisturbed ring samples are associated with identical textures is given in the MetaData table.

L131-132: To me the "accuracy….smaller than" structure limps. Revise? Uncertainty in their geo-position is less than 100m? Etc. Btw, is this true for all samples? If not, please state.
The sentence will be rephrased: "The geo-positions are reported with a lateral accuracy of 100 m…"

L143-144: Please list which methods those were. Was PARIO also involved?
We will provide a more detailed table of the respective analyses (as was suggested by referee #1, too).

L147: To my understanding using the "soiltexture" R-package means that essentially a log-linear interpolation. Is that correct? If so, I have to be critical of the approach. More advanced approached have already been used to re-classify European data more than 20 years ago. The key is to reduce resulting biases. If alternatives have been looked at, please justify why still this approach was to be used.

Yes, the "soiltexture" R-package only enables a log-linear interpolation. We agree with the raised concerns and we will revise the conversion method according to Nemes et al. (1999) https://doi.org/10.1016/S0016-7061(99)00014-2 and Minasny and McBratney (2001) https://doi.org/10.1071/SR00065. We will revisit the data and use a more sophisticated method if possible. However, since we provide much more detailed data within the German classification system, i) specific users can convert the data with their own tools and ii) the used methods are unlikely to result in large deviations.

L148-149: Re bulk density calculations: was the missing volume due to the earlier positioning of HYPROP tensiometers accounted for? Perhaps so. Please state for the record, so that others also think about it in the future.

The missing volume of the HYPROP tensiometers is 1 mL and is accounted for in the HYPROP software. Since this is part of the standard procedure, we do not consider it worthwhile to be noted here.

L169: near-saturated conditions.... (please also consider explaining in half a sentence why PDI enables that prediction better)

Although the dataset has originally been compiled for advances with the PDI (see Peters et al., 2021, https://doi.org/10.1029/2020WR029211, and Peters et al., 2023, https://doi.org/10.5194/hess-27-1565-2023) the SHP model is not the core topic of this manuscript. We will follow your suggestion and add a brief explanation and clear reference to the recently published paper.

L170-171: Sure, but please provide a very short summary of the method and the choice of -6cm.

For clarification, we will change the sentence to "To avoid an unrealistically sharp drop of the conductivity curve close to saturation, we constrained the conductivity model by a maximum pore radius (maximum tension) close to saturation with the "h-clip approach" of Iden et al. (2015). According to Jarvis (2007), the maximum tension was set to -6 cm (5 mm equivalent pore diameter)."
Jarvis (2007): https://doi.org/10.1111/j.1365-2389.2007.00915.x
Iden et al. (2015): https://doi.org/10.1016/j.advwatres.2015.09.005

L190: Has an SQL-supported, searchable single-file database format been considered?

Yes, but we have not seen any advantage in using SQL or any other higher level data storage format. We have a unique Sample-ID as key, so that users will be able to convert the .csv file data into any system. We find the simple file-based format as most flexible for this kind and amount of data.

L191: "Soil texture" is derived information, especially after interpolations. Is the raw particle-size data reported? In what format? How many points typically? It would be best practice to report it, so that future users can make their own choices of interpolation, as well as just have the more detailed particle-size data.

It is true that this is already interpreted/derived data after applying the Stokes law on sedimentation data and after recompilation with sieving and clay assessment. However, we simply did not propagate any more detailed "raw data" from the various labs. Since the central element of our manuscript are the soil hydraulic data which we provide in full detail, we consider the reported 7 soil texture classes as sufficiently informative reference.

L191: Still about sol texture: Is it reported in the database for which samples the original measurements were according to the German standard (silt at 63 microns) and for which those were according to the USDA/FAO standard (50 microns). Obviously the interpolation is for the other.
Pls. see the earlier reply to L101

L198: replace "contained" with "available"
Thank you. Will be replaced.

L201-204: Obviously there is large disparity in geographical distribution. My first gut feeling was: why not to limit the data to Germany? - but that would lead to loss of data. As an alternative, the authors could/should provide some information (data distribution, similarity in methodology, standards, use of particle-size interpolation (see above), etc.) that helps the eventual user decide against cutting off the Canadian, Japanese and Israeli data – citing methodological inhomogeneity - right away prior to running an analysis. For me, for instance, losing ca. 10% of the data in exchange for gaining more homogeneity seems like little cost to pay.
Pls. see the earlier reply to L85-89. We agree that the relatively few samples outside the central European context could easily be dropped. However, we actually hope that the data will motivate further contributions to a revised version, which are only now emerging with the highly standardized and comparable measurement technique. We thus opted to include the data and leave it to the users to select the data to their specific needs. Any user is free to either use or drop certain data for their specific usage.

L215: this is only true if we cut "natural soils" at the boundaries of temperate climate. Sandy clays and that region of the texture triange that is blank here are frequent in the tropics.
Thank you for your critical view. As outlined earlier, we agree to the geographical skewedness of the provided data to a central European temperate climate context. We will replace "natural" with "temperate climate".

L214-229 (section 3.2): This is a rather flat statistical summary that could be greatly shortened or just relied on in a small table. Instead, it would be much more useful to read about the handling of the raw particle-size data (why not include in the datadase?), interpolation (convince the user you chose the right method, provide which point was interpolated for how many samples that now carry extra uncertainty, etc.). I find the currently provided detail to be insufficient. You work with international data, and the particle-size conversion/harmonization aspect has been a bottle-neck in every one of such projects earlier (e.g. HYPRES, EU-HYDI) where data harmonization took place at all.
We acknowledge our shortcomings with respect to particle-size data. You are rightfully demanding more scrutiny and we will carefully check all data and available meta information to be more precise on this – from measurement to data harmonisation. However, we also have to clarify that one of the main features of the presented data is that soil texture classes have substantial limitations in their information about soil hydraulic properties. This does not relate

to texture classes in Fig 4 but to Fig 6. With our focus on the main advance of the dataset with combined retention and conductivity curve data of a large range of matric potential, we disagree that users will benefit from raw particle-size data. Most users from neighboring disciplines will probably reconsider their usage of standard van Genuchten - Mualem parameterizations. Some might find inspiration on how to complement our data with their own (and hopefully help to extend the dataset). Very specific users like you will demand for more details on particle size analyses and we hope to give a minimum reference through the methodological details we will add (see earlier replies on the matter). We believe that your argument is well-founded and would require additional analyses. Since most samples are retained, we invite you to complement our dataset with more advanced and harmonized particle size measurements.

L227-229: Definitely delete this. This is basically coded into the texture classes' definitions or even their names. As if one said that "sand content was higest in sands".
We believe this is a misunderstanding. We will revise the first sentence as follows: "In addition to the standard soil texture classification by sand, silt and clay fractions, the subgroups for silt and sand (i.e. coarse sand, medium sand, fine sand, coarse silt, medium silt, and fine silt) are also provided for the German classification system (Figure 3).

L241-243 and Figure 4: the circles, trianges and squares are only identifiable under extreme magnification. Please find another way of identifying them. Perhaps only refer to the pF ranges? Or colors?
You are certainly challenging the capabilities of plotting so many data points without any aggregation obscuring the main point of the figure... Since all plotted values have been quality checked and since their origin does not really make a huge difference in the figure's interpretation, we have ended up with this hard to discern version. However, we will follow your suggestion (in line with referee #1) and try our best to find a new version of these plots.

L248: (a) Please define what 'dry range' means. (b) What would be against suggesting that the mini-disk infiltrometer could be used for this in the laboratory?
You can find the definitions of the wet range (defined here as pF < 1.8; pF = $\log_{10}(-h \; [\text{cm}])$, medium range (1.8 < pF < 4.2) and dry range (pF > 4.2) in the manuscript in L134 and L140.
Using some sort of mini-disk (or better a miniature hood-) infiltrometer in the lab is certainly an interesting idea, which would also allow for retention and conductivity data on the wetting branch near saturation. However, this would require a development of a fully novel device with controlled boundary conditions and fluxes. Moreover, most tension infiltrometers will quickly reach their limits much earlier than pF 1. The original mini-disc infiltrometer ranges to pF 0.75 but can rarely be used so far. I have never reached more than pF 1.2 with a hood infiltrometer in the field. In most occasions pF 0.5 would be a very good value already. The limits are due to the air entry in the field, which could be partly controlled in the lab. But you should still consider the substantial difficulties, which might more relate back to pressure pot analyses than forward to advances with the Hyprop system. Despite this interesting idea, we think it is out of scope here, to deepen this discussion. Again, we would be pleased to continue this debate on another occasion.

L261-263: Can you please suggest why that is? Is it more realistic, or only a fall-out of model constraints?

As stated earlier, conductivity data are only available in a limited moisture range. Therefore, the conductivity is extrapolated in the wet and dry range. In the new PDI system (Peters et al., 2021; 2023), this extrapolation is done on a physical basis. We will add this information in the revised version.

L268: Why not just call them "commonly derived properties" …to evaluate the ability of a soil…..
Thank you. Will be revised as suggested.

L273-274: PAW is most often not the highest in the finest textured soils, but rather the intermediate to intermediately fine textured soils. Can you please refine the statement, or justify your stated finding?
Our statement was misleading since it only reported the ranges with erroneous reference to the texture. Fig 6c clearly points to the issue that PAW is not too well informed by texture alone and we will argue along this line in the revised version.

L288: saturation levels compared
L293: saturation levels
Thank you. Will be changed as suggested.

L295-298: This statement and follow-up elaboration does not make much sense in 2023. Most internationally used databases hold data _only_ of undisturbed samples. Some old ones have some disturbed ones. If you want to keep this statement, please give justice to the internationally known databases and cite which ones are based on disturbed and which are based on undisturbed samples. The statement here suggests as if this database is unique in this sense, whereas it is not.
We are not aware of any comparable dataset with undisturbed samples. Moreover we have presented some of the figures in talks and explicitly asked the audience to point us to comparable data from undisturbed samples without getting positive answers. Thus we would be pleased to get some more clarification on this.

L304: use comma before and after "similarly to Weynants et al. (2009)"
Will be done.

L307: Remove "Besides"
Will be addressed when revising the texture related aspects of the manuscript.

L308-309: This sounds like a statement from the 1990s. Please remove/revise/update according to the state of the art. (1) ALL "continuous" PTFs use particle size data, and not only classes or texture groups; (2) already in the late 90s some studies have evaluated the benefit from using finer-resolution particle-size data, as well as alternative representations of particle size distribution in PTFs (e.g. geometric mean and std of the curve), (3) the effect of using (and misusing) different classification systems was also evaluated at least 2 decades ago. You can use the Rawls and Pachepsky 2004 book as base reference, and find the most relevant literature therein.
We are under the impression that there is some discrepancy in the commonly used terminology and partly methods in the disciplinary subgroups and academic bubbles. We understand your concern raised throughout the manuscript and will carefully address it. We refer to the

commonly used PTFs. Again, a detailed review and discussion of PTFs is out of the scope of this data paper. We will slightly rephrase the sentence to:
"The effect of the resolution of particle size classification on SHP has rarely been investigated, and commonly used pedotransfer functions only consider the main texture groups as predictor variables"

L311: re: "more accurate PTFs": That always depends on the bottleneck in each source database and PTF development tool, as well as noises and biases in a database. Here I think there are at least two obvious data-bottlenecks: (1) not publishing the as-detailed-as-possible, original particle-size data (if that is the case), and (2) using a relatively weak interpolation technique to get from one classification system to the other. Otherwise yes, the data collection has good values.
Again thank you for pointing us to our weak formulation. We agree that there are data-bottlenecks and that we should not be part of this. However, there is also an information bottleneck, which lies outside the details of the particle size data and any interpolation. To detail on this, we will add a discussion section on exactly this after analyzing the potential effects and the remaining uncertainty.

L321: and provide
we will check the sentence again.

L322: handle such dependencies.
Thank you. Will be changed.

L325: which led to more
Thank you. Will be corrected..

L327-328: Sure, but that is easier said than done. It can also be addressed by using "local" PTF solutions instead of the usual "global" ones across the data domain. A local type PTF algorithms can work with dense data where it is dense within the overall domain, and scarce data where it is scarce. It is easy to set such techniques up to quantify that and communicate it together with the estimate, along with estimation uncertainty. The state-of-the-art has changed in the last 20 years.
Thank you. We will add the valuable information that locally calibrated PTFs might be useful for certain applications and will address this in the revised version.

L332: repeated from an earlier comment: what is against spelling out that the mini disk infiltrometer wold be suitable to respond to the need for "near saturated hydraulic conductivity"
And repeating from the reply: We agree that this could become interesting but would require a substantial revision of the Hyprop instrument and analytical procedure, which we are keen to debate but which we find outside the scope of this data manuscript.

L356: refine this to "compared to most data in existing databases", because there are actually evaporation-based data in some of the relatively newer databases like EU-HYDI, but even HYPRES has such already from the 1990s.
Thank you. Will be changed.

L368: delete "also"

Will be deleted.

Dear referee #2,
Thank you again for your critical assessment and the many suggestions. We hope that we could address all concerns and we understood them correctly. We will use your advice for a substantial revision of our manuscript.

Kind regards,
Tobias Hohenbrink, Conrad Jackisch, Wolfgang Durner, Kai Germer, Sascha Iden, Janis Kreiselmeier, Frederic Leuther, Johanna Metzger, Mahyar Naseri, and Andre Peters

---

## Author Response (AR1)

**Authors response to reviewer comments**

Dear Dr. Tian,

We are happy to submit a revised version of our manuscript. Following the suggestions by the referees we have implemented the following major, moderate and minor changes:

**Major changes**

1. We added a paragraph "Data preparation and quality check" as new section 2.4
2. We added an overview table about the measured variables and the methods used. We also provide more information about the methods.
3. We revised the texture class conversion estimate and used a more sophisticated and reliable approach to estimate the USDA sand/silt boundary at the particle size of 50 μm (Nemes et al., 1999). Additionally, we added three panels to Figure 2 showing the data in the USDA texture triangle. This will help international readers to interpret the soil texture information provided in the German classification system.
4. We discussed the methods used for texture analyses and possible improvements for future data collections in more detail.

**Moderate changes**

1. We have made a final check of the data and updated the data collection on the repository "GFZ Data Services".
2. We have provided more details about the PDI version used.
3. We carefully tested alternative plotting options for Figure 4. However, the current version in original print quality is still the best we can achieve. To guide the readers we amend the caption that the information is included in the online version after zooming in.
4. We added information about sampling depth to the MetaData table
5. We added information about methods of texture analysis to the MetaData table.
6. We added Figure 5 complementing Figure 4 with the effect of bulk density on soil hydraulic properties and discussed the predictive power of soil texture and bulk density on soil hydraulic properties.

**Minor changes**

We implemented all minor correction suggestions (e.g. spelling and grammar errors, precise technical terms, etc.) directly.

**Details as replies to the two referees**

Ma#: major changes (# number of list above)
Mo#: moderate changes (# number of list above)
Mi: minor changes (not listed above)
Answer without changes: Arguments only stated in reply letter

**Reply to Referee #1**

GENERAL COMMENTS
The presented dataset stores high quality soil physical data. The description of measurement methods and models applied to compute soil hydraulic parameters by fitting the moisture retention and hydraulic conductivity are detailed and clear. Structure of the manuscript is logical. The main strength of the database is the data on unsaturated hydraulic conductivity. This way the presented work and dataset will attain international interest.
The data could be easily accessed. Organization of the six data tables within the dataset is logical, the tables can be merged by the Sample_ID column.

Dear referee #1,
Thank you very much for your positive assessment and the thoughtful suggestions for a revision of our manuscript. We will address these line by line in the following.

A paragraph could be added about data quality check under materials and methods, because that could strengthen that the dataset was rigorously checked and the way the check was performed can be very informative for the readers and serve as a guideline.

**Ma1:** You are perfectly right that such a quality check paragraph is a useful addition. It got lost during our initial internal revisions. We have added a respective section (new 2.4 ) describing the quality assessment of the data in more detail.

A final data check would be useful to secure that all data is correct. The detailed review can be found under SPECIFIC COMMENTS.

**Mo1:** The data has been checked again.

SPECIFIC COMMENTS
L24-25, L55, L78, L79, L82, L96 and entire text, please specify if you refer to soil profiles or soil samples, the word "data sets" is not enough specific.
**Mi:** We generally refer to soil samples throughout the manuscript. We have specified this more clearly in e.g. L24-27, L100 and L 127-137 of the revised manuscript.

L101-102: please add reference or some examples for the two level texture information, because it is not widely used.
**Mi:** We thank the reviewer for this hint. We have added references of the soil classification systems to the sentence. More information about the German sub-classes of soil texture is provided in Chapter 3.2 and Figure 3a.

L127: … mixed average soil sample … is it correct?
**Mi:** Yes, the formulation was misleading. We have amended the paragraph accordingly.

L134-149: all is clearly described, just a table providing an overview about the methods would be very informative, because for the readers it is a very valuable information what method was used for which soil property. Please add information about the measurement method of N and S, as well – because those are also included in the BasicProp.csv file. Please consider if the method used by soiltexture package can have limitations. Some other methods exist, which might result in a more accurate conversion to USDA silt and sand content. It is possible that in your case there would not be significant difference between different methods, but for other cases there might be. Readers might follow the procedure you published, so it worth to mention other methods, e.g.: Nemes et al (1999) https://doi.org/10.1016/S0016-7061(99)00014-2.
**Ma2:** We now provide the information as new  Table 2.
**Ma3:** We followed your suggestion and used Nemes et al. 1999 to convert the soil texture systematics.

L150: Before "2.4 Fitting models to measured data" subsection could you please add a separate subsection on how quality of the data was secured? Could you shortly describe what rules were applied during checking the data?
**Ma1:** Done as new section 2.4

L181-182: please add reference and equation used to compute parameter Ks of the PDI model.
**Mo2:** The calculation of Ks was done as described by Peters et al. (2023), https://doi.org/10.5194/hess-27-1565-2023. This was made clearer in the revised manuscript. Since this calculation procedure is rather complicated, we would like to restrain from a repetition of the procedure in this paper.

L184-186, Table 1: please add meaning of VGM and PDI to have the table self explanatory.
**Mi:** The caption has been revised.

L190-193: would be informative to add 4.1-4.3 tables from 2023-012_Hohenbrink-et-al_Data-Description.pdf file here.
**Mi:** We have added table 2 as overview about the variables and methods. The data tables are more suitable in the data description readme.

L194: It might worth to consider to create a metadata .xml file following the INSPIRE metadata guidelines (ISO 19115 and ISO 19139) and add it to the dataset.
**Mi:** We are aware of the different metadata guidelines and fully support the notion to emphasize their implementations. We have double checked if an xml file following the INSPIRE or other templates is suitable. However, we believe that the geographic references are fully traceable and that the provided tabled metadata is much more accessible for potential users.

L216-219 and Figure 2. : please consider to provide this information according to USDA texture classes (based on the USDA sand, silt and clay fractions), because that is internationally used, the German texture classes are not widely known out of Germany. I see that for Figure 3. it might not make sense to use the USDA standard because than you might have only three fractions and Figures 4 and 5 is easier to interpret if meaning of texture classes can be read from Figure 3.
**Ma3:** We have done the reclassification and provide the USDA texture classes in Figure 2.

L241: circles on Figure 4 are hardly visible, maybe Figure 4 could be edited somehow to let easier distinguish between circle, triangle and square.
**Mo3:** We have not succeeded in improving this plot. Thus we have amended the caption accordingly.

L244: Please shortly add why number of dewpoint measurements ranges between 1 and 8.
**Mi:** The statement has been amended.

L263: … range for coarser texture classes … Do you agree?
**Mi:** We have rephrased the two sentences to clarify the formerly misleading statement.

L268-271: if th_1_8, th_2_5 and th_4_2 columns of Param table were computed with PDI model, please add "_PDI" as last characters to those column names.
**Mi:** Thank you for pointing to this. We have amended the column names accordingly.

L272-273: please add very short explanation for why filed capacity and wilting point vary widely within texture classes. This is obvious for experts in soil physics but not that trivial for researchers from other environmental fields.
**Mi:** We added: "which directly results from the variation of the retention curves within a single texture class (Figure 5c)"

L308: please consider e.g. the work of Twarakavi et al. (2010) (https://doi.org/10.1029/2009WR007939 ) - or possible other papers in this topic – and rephrase the sentence accordingly.

**Mo6:** Thank you for suggesting this citation and challenging our sloppy formulation. We have adapted our discussion to become more clear in this regard and included the work of Twarakawi et al. (2010).

L311: Do authors plan to add soil depth, chemical soil properties - e.g. pH or calcium-carbonate content - or taxonomical information to the dataset in the future? If soil depth is available it might be easy to add to the BasicProp.csv table, it could be an important data column.

**Mo4:** We have added a column with sampling depths to the MetaData.csv. Reliable data about pH or calcium-carbonate content is unfortunately not available.

Result of checking the database:
there is a negative theta value in RetMeas.csv, please check and revise/correct.
there is a negative value for S in BasicProp.csv, please check and revise/correct.
Sum of USDA sand and silt and clay is 99.9 and 100.1 for some samples, it might worth to correct them to sum up to 100.

**Mo1:** Thank you for pointing to these issues. We have checked the data again and updated them on the repository.

Dear referee #1,
Thank you again for your suggestions. We hope that we could address all concerns. We have used your advice for a substantial revision of our manuscript and a check of the data.

Kind regards,
Tobias Hohenbrink, Conrad Jackisch, Wolfgang Durner, Kai Germer, Sascha Iden, Janis Kreiselmeier, Frederic Leuther, Johanna Metzger, Mahyar Naseri, and Andre Peters

**Reply to Referee #2**

The manuscript reports on a decently homogeneous data collection of soil physical and hydraulic properties that, in many aspects, provides more detailed data than what is available in most existing and internationally available databases. However, in certain aspects it provides less information, or potentially uses a weak solution for data harmonization. If unresolved, these can become its limitations, and eventually limit the anticipated benefit from reporting very detailed water retention data. This contribution of data is very welcome in the literature, but it would be desirable to report the data and some of the methods in more detail.

Dear referee #2,

Thank you very much for your critical assessment and the thoughtful suggestions for a revision of our manuscript. We will address these line by line in the following.

**Answer without changes:** As a preamble to the forthcoming replies we are under the impression that some sort of misunderstanding underlies some of the critical comments. We seek to convey the combined soil water retention and hydraulic conductivity curve data as a valuable basis for a series of applications (including SHP model and PTF development among others). The additional data (e.g. organic carbon contents) has been added for further reference and as a "service" for a broader usability of the data. While your comments, suggestions and discussion aspects provide valuable hints for our revision, we also want to stick to our original plan for the publication and explanation of exactly this data. We will certainly elaborate more clearly on the respective methods, but we cannot perform additional analyses with respect to more texture data. Although we have changed the conversion of the

texture classes, we are, of course, bound to the laboratory data we have. And again, we regard this data as additional information complementing the characteristic curves.

**Answer without changes:** We sincerely hope that we have well understood your raised concerns and that our replies are capable of conveying our gratitude for your constructive criticism.

Three generic and a number of specific comments follow.
Some data, primarily particle size distribution data should be reported in more detail. I don't see trail of reporting more than sand-silt-clay contents, whereas the original set of measurements should be reported. There are emerging directions of research that would utilize that. It should also be communicated which of the silt-sand data pairs are from original measurements, and which have been a product of interpolation that adds additional noise.
**Answer without changes:** We appreciate the suggestion. We focus on soil water retention and hydraulic conductivity curve data. We have tried to harmonize the data as much as possible and we did not anticipate such interest in these "auxiliary" data. Based on your comments, we have revised the data assemblage accordingly. However, unlike the highly standardized Ksat, Hyprop, WP4C analysis procedure, the texture analysis slightly differs between the originating labs and even for specific samples. Reporting the original set of measurements (as weight of retained sediment during wet sieving and sediment aspiration during sedimentation experiments including lab temperature etc.) is outside the scope of this publication and too much manual work to retrieve from the respective lab books.

In addition, instead of the rather standard, flat data description, it is better value to go in depth on the exact steps that involved data manipulation, and present the outcome in a convincing way. I specifically refer to the particle-size conversions and its uncertainty, as well as the derivation of bulk density and return to these in the detailed comments section.
**Ma2:** In line with referee #1 we agree that this part got shortened too much. Thank you for your comment. We have revised and extended the degree of detail in our methodological descriptions accordingly. However, we do not expect the main characteristics of the data to be dependent on the precision of the particle-size conversion. We report the texture data in 7 classes (German standards) all based on wet sieving and sedimentation analyses.
**Mo5:** We added information about methods of texture analysis to the MetaData table.
**Mo6:** In figures 4, 5, 6 we demonstrate that the information about the hydraulic functioning (retention and conductivity curves, PAW) is only weakly encoded in the texture information alone. This has been shown earlier too and underpins the main reason for us reporting the continuously measured unsaturated hydraulic conductivity in conjunction with soil water retention (although limited by methodological constraints).

Several statements made would have been true in the 1990s, but not anymore. I highlight some among the detailed comments. The authors should revise those and bring the statements up to standard according to the state-of-the-art in the 2020s.
**Answer without changes:** We agree that we mostly refer to consolidated concepts in soil physics and soil analyses as we intend the dataset to be hopefully useful for a broader range of neighboring disciplines. We are under the impression that there is some misunderstanding due to awkward wordings on our side. We consider the SHP model PDI (as just published in a new version, Peters et al. 2023, https://doi.org/10.5194/hess-27-1565-2023) as up to date. With respect to the discussed usage of the data for PTF development, we refer to algorithms predicting SHP model parameters from more commonly accessible information in a continuous manner. Such PTF are driven by information about texture classes, bulk density and soil organic content.
**Ma3, Ma4, Mo5, Mo6:** Although we think that our arguments are not completely outdated, we have carefully revised them. Moreover we have clarified our wording to avoid any misunderstanding in the revised version.

Specific comments:
Abstract: Include a brief reference to the geographical extent (i.e. the contributing list of countries, with Germany dominating)

**Mi:** We added "mostly central European".

L63: I believe Brazil has non-tropical data in HYBRAS as well. Please check and remove the word 'tropical' if necessary.
**Mi:** This is true and unnecessary. Tropical has been removed.

L66: replace 'commonly' with 'openly'
**Mi:** Has been replaced.

L72: …as it has been often done historically.
**Mi:** Has been added.

L75: … which are often not recorded at the time of sampling.
**Mi:** "which are often not recorded nor assessed at the time of sampling" has been added.

L85-89: IT has been identified that ROSETTA's data is also geographically skewed. It would be worth exploring where the 235 samples with unsatK come from.
**Answer without changes:** Geographical skewness is an issue. However it is an issue which we hope to address in the future when we succeed in promoting the value of standardized analyses and evaluation as presented with the soil hydraulic property data. A review of the spatial references in the literature and its potential skewness is beyond the scope of our data manuscript.

L99: i.e. the evaporation
**Mi:** Thank you. It has been changed.

L99: dew-point potentiometry (Campbell et al., 2007)
**Mi:** Thank you. Revised.

L101: these are not two 'levels', but two standards. Use e.g.: "provided according to both the German and the USDA classification systems, and the…"
**Mi:** Thank you. Our wording was unclear. We have revised the sentence to "Soil texture information is provided according to both the German (Ad-hoc-Arbeitsgruppe Boden, 2005) and the USDA classification systems (USDA, 1999). Within the silt and sand classes, we also provide the sub-classes coarse, medium and fine according to the German system."

L103: please avoid using text like "strong foundation". The users and history will decide that.
L104-105: delete this sentence, it is repeated from earlier. Remove the dependence of the next sentence on this sentence (reference to further purposes).
L109: This is the 3rd mention of "various original purposes". Perhaps remove both earlier mentions.
**Mi:** All true. The statements have been shortened and rephrased.

L129: here you mean aggregated DISTURBED samples, is the correct?
**Mi:** Yes, the formulation was erroneous. Thank you for pointing this out. DISTURBED samples (originating from the undisturbed ones or having been sampled alongside the undisturbed samples) have been analyzed. In some cases, the disturbed reference samples have been referring to several undisturbed ring samples. In such cases, the data were averaged and attributed to all rings. We have corrected the formulation accordingly. Information about which undisturbed ring samples are associated with identical textures is given in the MetaData table.

L131-132: To me the "accuracy….smaller than" structure limps. Revise? Uncertainty in their geo-position is less than 100m? Etc. Btw, is this true for all samples? If not, please state.
**Mi:** The sentence has been rephrased: "The geo-positions are reported with a lateral accuracy of 100 m…"

L143-144: Please list which methods those were. Was PARIO also involved?

**Ma2, Mo5:** We added an overview about the measured variables and the methods used (Table 2). We also added a column to the MetaData.csv with the methods of texture analysis used for individual samples.

L147: To my understanding using the "soiltexture" R-package means that essentially a log-linear interpolation. Is that correct? If so, I have to be critical of the approach. More advanced approaches have already been used to re-classify European data more than 20 years ago. The key is to reduce resulting biases. If alternatives have been looked at, please justify why still this approach was to be used.

**Ma3:** Yes, the "soiltexture" R-package only enables a log-linear interpolation. We agree with the raised concerns. In the revised version ve have converted the texture data by interpolation with monotone cubic splines fitted to the cumulative particle size distributions as recommended by Nemes et al. (1999).

L148-149: Re bulk density calculations: was the missing volume due to the earlier positioning of HYPROP tensiometers accounted for? Perhaps so. Please state for the record, so that others also think about it in the future.

**Answer without changes:** The missing volume of the HYPROP tensiometers is 1 mL and is accounted for in the HYPROP software. Since this is part of the standard procedure, we do not consider it worthwhile to be noted here.

L169: near-saturated conditions…. (please also consider explaining in half a sentence why PDI enables that prediction better)

**Mo2:** Thank you for that suggestion. We changed the passage to: "Unlike VGM and common models of SHP, where the relative hydraulic conductivity curve is scaled by the saturated conductivity Ks, the new PDI model structure allows a physically based absolute conductivity prediction. Since conductivity data close to saturation are usually not available, this scheme enables a more realistic conductivity prediction under nearly saturated conditions (Peters et al., 2023)."

L170-171: Sure, but please provide a very short summary of the method and the choice of -6cm.

**Mo2:** For clarification, we have changed the sentence to "To avoid an unrealistically sharp drop of the conductivity curve close to saturation, we constrained the conductivity model by a maximum pore radius (maximum tension) close to saturation with the "h-clip approach" of Iden et al. (2015). According to Jarvis (2007), the maximum tension was set to -6 cm (5 mm equivalent pore diameter)."
Jarvis (2007): https://doi.org/10.1111/j.1365-2389.2007.00915.x
Iden et al. (2015): https://doi.org/10.1016/j.advwatres.2015.09.005

L190: Has an SQL-supported, searchable single-file database format been considered?

**Answer without changes:** Yes, but we have not seen any advantage in using SQL or any other higher level data storage format. We have a unique Sample-ID as key, so that users will be able to convert the .csv file data into any system. We find the simple file-based format as most flexible for this kind and amount of data.

L191: "Soil texture" is derived information, especially after interpolations. Is the raw particle-size data reported? In what format? How many points typically? It would be best practice to report it, so that future users can make their own choices of interpolation, as well as just have the more detailed particle-size data.

**Answer without changes:** It is true that this is already interpreted/derived data after applying the Stokes law on sedimentation data and after recompilation with sieving and clay assessment. However, we simply did not propagate any more detailed "raw data" from the various labs. Since the central elements of our manuscript are the soil hydraulic data which we provide in full detail, we consider the reported 7 soil texture classes as sufficiently informative reference.

L191: Still about sol texture: Is it reported in the database for which samples the original measurements were according to the German standard (silt at 63 microns) and for which those were according to the USDA/FAO standard (50 microns). Obviously the interpolation is for the other.
**Answer without changes:** Pls. see the earlier reply to L101

L198: replace "contained" with "available"
**Mi:** Thank you. Has been rephrased.

L201-204: Obviously there is large disparity in geographical distribution. My first gut feeling was: why not to limit the data to Germany? - but that would lead to loss of data. As an alternative, the authors could/should provide some information (data distribution, similarity in methodology, standards, use of particle-size interpolation (see above), etc.) that helps the eventual user decide against cutting off the Canadian, Japanese and Israeli data – citing methodological inhomogeneity - right away prior to running an analysis. For me, for instance, losing ca. 10% of the data in exchange for gaining more homogeneity seems like little cost to pay.
**Answer without changes:** Pls. see the earlier reply to L85-89. We agree that the relatively few samples outside the central European context could easily be dropped. However, we actually hope that the data will motivate further contributions to a revised version, which are only now emerging with the highly standardized and comparable measurement technique. We thus opted to include the data and leave it to the users to select the data to their specific needs. Any user is free to either use or drop certain data for their specific usage.

L215: this is only true if we cut "natural soils" at the boundaries of temperate climate. Sandy clays and that region of the texture triangle that is blank here are frequent in the tropics.
**Mi:** Thank you for your critical view. As outlined earlier, we agree to the geographical skewedness of the provided data to a central European temperate climate context. We have replaced "natural" with "temperate climate".

L214-229 (section 3.2): This is a rather flat statistical summary that could be greatly shortened or just relied on in a small table. Instead, it would be much more useful to read about the handling of the raw particle-size data (why not include in the datadase?), interpolation (convince the user you chose the right method, provide which point was interpolated for how many samples that now carry extra uncertainty, etc.). I find the currently provided detail to be insufficient. You work with international data, and the particle-size conversion/harmonization aspect has been a bottle-neck in every one of such projects earlier (e.g. HYPRES, EU-HYDI) where data harmonization took place at all.
**Ma3, Ma4, Mo5, Mo6:** We acknowledge our shortcomings with respect to particle-size data. You are rightfully demanding more scrutiny and we will carefully check all data and available meta information to be more precise on this – from measurement to data harmonisation.
**Answer without changes:** However, we also have to clarify that one of the main features of the presented data is that soil texture classes have substantial limitations in their information about soil hydraulic properties. This does not relate to texture classes in Fig 3 but to Fig 6. With our focus on the main advance of the dataset with combined retention and conductivity curve data of a large range of matric potential, we disagree that users will benefit from raw particle-size data. Most users from neighboring disciplines will probably reconsider their usage of standard van Genuchten - Mualem parameterizations. Some might find inspiration on how to complement our data with their own (and hopefully help to extend the dataset). Very specific users like you will demand for more details on particle size analyses and we hope to give a minimum reference through the methodological details we will add (see earlier replies on the matter). We believe that your argument is well-founded and would require additional analyses. Since most samples are retained, we invite you to complement our dataset with more advanced and harmonized particle size measurements.

L227-229: Definitely delete this. This is basically coded into the texture classes' definitions or even their names. As if one said that "sand content was higest in sands".
**Mi:** We believe this is a misunderstanding. We have rephrased the first sentence as follows:"In addition to the standard soil texture classification by sand, silt and clay fractions, the subgroups for silt

and sand (i.e. coarse sand, medium sand, fine sand, coarse silt, medium silt, and fine silt) are also provided for the German classification system (Figure2).

L241-243 and Figure 4: the circles, trianges and squares are only identifiable under extreme magnification. Please find another way of identifying them. Perhaps only refer to the pF ranges? Or colors?

**Mo3:** We have not succeeded in improving this plot. Thus we have amended the caption accordingly.

L248: (a) Please define what 'dry range' means. (b) What would be against suggesting that the mini-disk infiltrometer could be used for this in the laboratory?

**Answer without changes:** You can find the definitions of the wet range (defined here as pF < 1.8; pF = $\log_{10}$( -$h$ [cm]), medium range (1.8 < pF < 4.2) and dry range (pF > 4.2) in the manuscript in the first paragraph of chapter 2.3 "Laboratory measurements".

**Answer without changes:** Using some sort of mini-disk (or better a miniature hood-) infiltrometer in the lab is certainly an interesting idea, which would also allow for retention and conductivity data on the wetting branch near saturation. However, this would require a development of a fully novel device with controlled boundary conditions and fluxes. Moreover, most tension infiltrometers will quickly reach their limits much earlier than pF 1. The original mini-disc infiltrometer ranges to pF 0.75 but can rarely be used so far. I have never reached more than pF 1.2 with a hood infiltrometer in the field. In most occasions pF 0.5 would be a very good value already. The limits are due to the air entry in the field, which could be partly controlled in the lab. But you should still consider the substantial difficulties, which might more relate back to pressure pot analyses than forward to advances with the Hyprop system. Despite this interesting idea, we think it is out of scope here, to deepen this discussion. Again, we would be pleased to continue this debate on another occasion.

L261-263: Can you please suggest why that is? Is it more realistic, or only a fall-out of model constraints?

**Mo2:** As stated earlier, conductivity data are only available in a limited moisture range. Therefore, the conductivity is extrapolated in the wet and dry range. In the new PDI system (Peters et al., 2021; 2023), this extrapolation is done on a physical basis. We have added this information in the revised version.

L268: Why not just call them "commonly derived properties" …to evaluate the ability of a soil…..

**Mi:** Thank you. We have changed the sentence.

L273-274: PAW is most often not the highest in the finest textured soils, but rather the intermediate to intermediately fine textured soils. Can you please refine the statement, or justify your stated finding?

**Mi related to Mo6:** We have changed the sentence to: "Plant available water content (Figure 6c) depicts the same high variability within the texture triangle. It varies between the extremes of 3.8 vol. % in pure sand up to 49.2 vol. % in fine-textured soil but does not align to any clear, texture-related pattern."

L288: saturation levels compared
L293: saturation levels
**Mi:** Thank you. It has been changed as suggested.

L295-298: This statement and follow-up elaboration does not make much sense in 2023. Most internationally used databases hold data _only_ of undisturbed samples. Some old ones have some disturbed ones. If you want to keep this statement, please give justice to the internationally known databases and cite which ones are based on disturbed and which are based on undisturbed samples. The statement here suggests as if this database is unique in this sense, whereas it is not.

**Answer without changes:** We are not aware of any comparable dataset with undisturbed samples. Moreover we have presented some of the figures in talks and explicitly asked the audience to point us to comparable data from undisturbed samples without getting positive answers. Thus we would be pleased to get some more clarification on this.

L304: use comma before and after "similarly to Weynants et al. (2009)"
**Mi:** Has been changed.

L307: Remove "Besides"
**Mi:** Has been changed.

L308-309: This sounds like a statement from the 1990s. Please remove/revise/update according to the state of the art. (1) ALL "continuous" PTFs use particle size data, and not only classes or texture groups; (2) already in the late 90s some studies have evaluated the benefit from using finer-resolution particle-size data, as well as alternative representations of particle size distribution in PTFs (e.g. geometric mean and std of the curve), (3) the effect of using (and misusing) different classification systems was also evaluated at least 2 decades ago. You can use the Rawls and Pachepsky 2004 book as base reference, and find the most relevant literature therein.
**Mi:** We are under the impression that there is some discrepancy in the commonly used terminology and partly methods in the disciplinary subgroups and academic bubbles. We understand your concern raised throughout the manuscript and have carefully addressed it. We refer to the commonly used PTFs. Again, a detailed review and discussion of PTFs is out of the scope of this data paper. We have revised the paragraph and now discuss the predictive power of soil texture and bulk density on soil hydraulic properties in more detail.

L311: re: "more accurate PTFs": That always depends on the bottleneck in each source database and PTF development tool, as well as noises and biases in a database. Here I think there are at least two obvious data-bottlenecks: (1) not publishing the as-detailed-as-possible, original particle-size data (if that is the case), and (2) using a relatively weak interpolation technique to get from one classification system to the other. Otherwise yes, the data collection has good values.
**Ma3, Ma4, Mo5, Mo6:** Again thank you for pointing us to our weak formulation. We agree that there are data-bottlenecks and that we should not be part of this. However, there is also an information bottleneck, which lies outside the details of the particle size data and any interpolation. To detail on this, we added a discussion about the predictive power of soil texture and bulk density on soil hydraulic properties.

L321: and provide
**Mi:** Has been changed.

L322: handle such dependencies.
**Mi:** Thank you. Has been changed.

L325: which led to more
**Mi:** Thank you. Has been changed.

L327-328: Sure, but that is easier said than done. It can also be addressed by using "local" PTF solutions instead of the usual "global" ones across the data domain. A local type PTF algorithms can work with dense data where it is dense within the overall domain, and scarce data where it is scarce. It is easy to set such techniques up to quantify that and communicate it together with the estimate, along with estimation uncertainty. The state-of-the-art has changed in the last 20 years.
**Mi:** Thank you. We added ", but this is a major task at the level of the soil hydrological community and can hardly be achieved by individual researchers."
We have decided not to go into detail about the local type PTF algorithms, as this would go too far in this paper, which is only intended to present the data.

L332: repeated from an earlier comment: what is against spelling out that the mini disk infiltrometer wold be suitable to respond to the need for "near saturated hydraulic conductivity"
**Answer without changes:** And repeating from the reply: We agree that this could become interesting but would require a substantial revision of the Hyprop instrument and analytical procedure, which we are keen to debate but which we find outside the scope of this data manuscript.

L356: refine this to "compared to most data in existing databases", because there are actually evaporation-based data in some of the relatively newer databases like EU-HYDI, but even HYPRES has such already from the 1990s.
**Mi:** Thank you. Will changed it.

L368: delete "also"
**Mi:** Has been deleted.

Dear referee #2,
Thank you again for your critical assessment and the many suggestions. We hope that we could address all concerns and we understood them correctly. We have used your advice for a substantial revision of our manuscript.

Kind regards,
Tobias Hohenbrink, Conrad Jackisch, Wolfgang Durner, Kai Germer, Sascha Iden, Janis Kreiselmeier, Frederic Leuther, Johanna Metzger, Mahyar Naseri, and Andre Peters

---

## Author Response (AR2)

**Authors response to reviewer comments**

Dear Dr. Tian,
Dear referees,

Thank you for helping us to improve our manuscript with your comments. Below you will find our responses to the last three comments. We hope that all open questions are now answered and that the paper is ready for publication.

Best Regards,
Tobias Hohenbrink, Conrad Jackisch, Wolfgang Durner, Kai Germer, Sascha Iden, Janis Kreiselmeier, Frederic Leuther, Johanna Metzger, Mahyar Naseri, and Andre Peters

**Review report #1:**
The authors addressed all the topics raised during the review. Only the following minor issues could be solved before publication:

1. present Table 2 is very informative, maybe reference to HYPROP, falling and constant head method, BD measurement method, wet sieving, hydrometer method, high-temp. combustion could be added,
Answer: We have added references for the methods or references to the measuring devices used.

2. L259: Figure 1: … USDA (d-f) system … ,
Answer: It has been corrected.

3. L294-295: the data set was not available from the link given in the manuscript, therefore it was not possible to check changes added to the dataset, e.g.: adding information on soil depth, modifying some column names, revising negative values; this is the reason for indicating data quality "Good" and not "Excellent".
Answer: We have implemented all the proposed changes to the data and their technical description on the repository. The revised data set can currently be accessed via the following review link:
https://dataservices.gfz-potsdam.de/panmetaworks/review/5c617cd2664ea4d03e81301b5bc2236f1948a3cf7eb9bad48da940524f0cbac0/
We assume that the link was corrupted during PDF conversion of the manuscript and would like to apologize for this.

---

## Author Response (AR3)

Dear Dr. Tian,

We are very pleased that our manuscript has been accepted for publication in ESSD. We thank you very much for handling the manuscript.

Best Regards,
Tobias Hohenbrink, Conrad Jackisch, Wolfgang Durner, Kai Germer, Sascha Iden, Janis Kreiselmeier, Frederic Leuther, Johanna Metzger, Mahyar Naseri, and Andre Peters